# SimDiffPDE: Simple Diffusion Baselines for Solving Partial Differential Equations

## Abstract

We showcase good capabilities of the plain diffusion model with Transformers (SimDiffPDE) for general partial differential equations (PDEs) solving from various aspects, namely simplicity in model structure, scalability in model size, flexibility in training paradigm, and universality between different PDEs. Specifically, SimDiffPDE reformulates PDE-solving problems as the image-to-image translation problem, and employs plain and non-hierarchical diffusion model with Transformer to generate the solutions conditioned on the initial states/parameters of PDEs. We further propose a multi-scale noise to explicitly guide the diffusion model in capturing information of different frequencies within the solution domain of PDEs. SimDiffPDE achieves a remarkable improvement of **+51.4%** on the challenging Navier-Stokes equations. In benchmark tests for solving PDEs, such as Darcy Flow, Airfoil, and Pipe for fluid dynamics, as well as Plasticity and Elasticity for solid mechanics, our SimDiffPDE-B achieves significant relative improvements of **+21.1%**, **+11.3%**, **+15.2%**, **+25.0%**, and **+23.4%**, respectively. Models and code shall be released upon acceptance.

## 1 Introduction

Solving partial differential equations (PDEs) is immensely important in extensive real-world applications, such as weather forecasting (Pathak et al., 2022; Chen et al., 2023; Bi et al., 2023), industrial design (Sekar et al., 2019; Jing et al., 2022; Liu et al., 2024), and material analysis (Roubíček, 2013; Kadic et al., 2019). As a basic scientific problem, it is usually difficult to obtain analytic solutions for PDEs. Therefore, the solutions of PDEs are typically discretized into meshes and then solved by numerical methods (Rodi, 1997; Zhao, 2008; Greenfeld et al., 2019), which usually takes a few hours or even days for complex structures (Umetani & Bickel, 2018). To deal with these issues, there has recently been rapid progress in deep learning-based methods (Li et al., 2020; 2024b; Lu et al., 2021), which typically tackles the challenging task using convolutional neural networks or transformers. Thanks to their impressive nonlinear modeling capacity, they can learn to approximate the input and output mappings of PDE-governed tasks from data during training and then infer the solution significantly faster than numerical methods (Goswami et al., 2022; Wu et al., 2023).

To date, major deep-learning-based methods can be broadly classified into three categories: (1) neural approaches that approximate the solution function of the underlying PDE (Han et al., 2018; Raissi et al., 2019); (ii) hybrid approaches (Arcomano et al., 2022; Bar-Sinai et al., 2019; Berthelot et al., 2023; Greenfeld et al., 2019; Kochkov et al., 2021; Sun et al., 2023), where neural networks either augment numerical solvers or replace plats of them; (iii) neural approaches in which the learned evolution operator iteratively maps the current approximate solution to a future state of the approximate solution (Bhatnagar et al., 2019; Brandstetter et al., 2022a;b). Despite that approaches (i) have achieved great success in modeling inverse and high-dimensional problems, and approaches (ii-iii) have started advance fluid and weather modeling in two and three dimensions, these methods typically learn a *deterministic* mapping between input coefficients and their solutions. However, due to the chaotic nature of some dynamics system described by PDE, *e.g.,* Navier-Stokes equation, even small ambiguities of the spatially averaged states as the inputs can lead to fundamentally different solutions over time, which leads the *deterministic* methods providing non-robust answers.

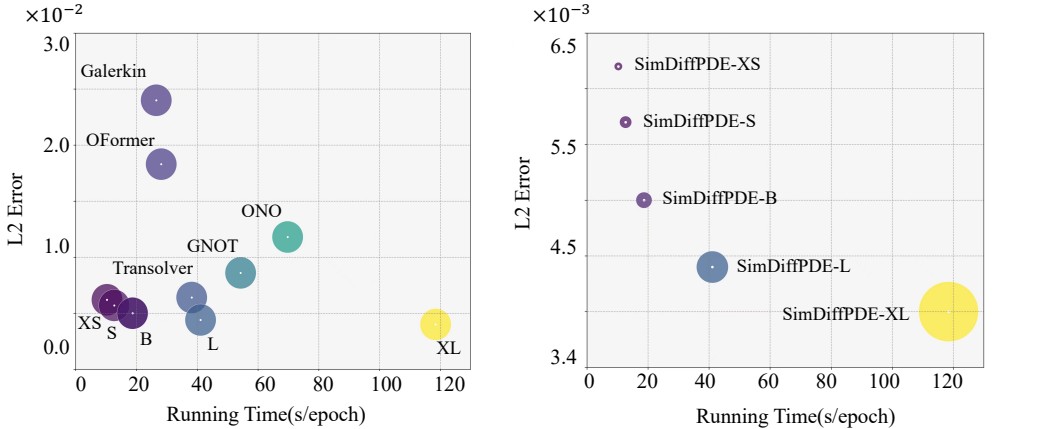

Figure 1: Left: Comparison of model performance across different benchmarks. XS: XS-SimDiffPDE, S: S-SimDiffPDE, B: B-SimDiffPDE, L: L-SimDiffPDE, and XL: XL-SimDiffPDE. Right: Comparison of model performance across different model sizes.

In comparison, *generative* diffusion models (Rombach et al., 2022) offer substantial potential for solving PDEs, especially those describing highly nonlinear systems, exhibiting capabilities similar to those in video prediction based on initial frames and auxiliary conditioning. Specifically, diffusion models can construct generative distributions that closely approximate the underlying probabilistic solution distributions instead of one solution point. Therefore, by ensembling solutions by sampling difference Guassian noise as inputs during the inference phase, diffusion models can produce more robust and accurate solutions of PDEs that particularly describe nonlinear and even chaotic systems.

In this paper, we demonstrate that plain diffusion models can be repurposed as effective and general PDE solvers (SimDiffPDE), with the multi-scale noise. The key to unlocking the potential of diffusion models lies in their ability to efficiently capture patterns of multiple scales in the solution domain. However, we observe that the default Guassin noise can not efficiently destroy the large-scale pattern in the *forward process*, and therefore the diffusion model can not learn to recover the large-scale pattern efficiently in the *reverse process*. By adding multi-scale noise in the *forward process*, the diffusion models are more explicitly required to learn to denoise the multi-scale noise to reconstruct multi-scale patterns of PDE solutions. During the inference phase, we leverage the test-time ensemble method to consider the generated solution distributions by sampling multiple Guassian noises as inputs. The two designs not only maintains structural simplicity but also significantly improves accuracy and robustness compared to previous state-of-the-art solvers. Our model consistently surpasses previous state-of-the-art models across six benchmarks involving various types of PDEs (Wu et al., 2024; 2022; Li et al., 2022b; Hao et al., 2023; Xiao et al., 2023). Notably, we achieve a **+51.4%** improvement in the challenging Navier-Stokes equations. For benchmarks for solving partial differential equations, *e.g.,* Darcy Flow, Airfoil and Pipe that describe fluids, Plasticity and Elasticity that describe solids, our SimDiffPDE-B achieves considerable relative improvements of **+21.1%**, **+11.3%**, **+15.2%**, **+25.0%**, and **+23.4%**, respectively.

Besides the superior performance, we also show the surprisingly good capabilities of SimDiffPDE from various aspects, namely simplicity, scalability, flexibility and universality. 1) For simplicity, due to the strong generative feature representation ability, the SimDiffPDE framework is rather simple. For example, it does not require any specific domain knowledge for architecture design and enjoys a plain and non-hierarchical structures by simply stacking several diffusion transformer layers. 2) The simplicity in structure brings the excellent scalability properties of SimDiffPDE. To be more specific, one can easily control the model size by stacking different number of diffusion transformer layers and increase or decrease feature dimensions, *e.g.,* we design SimDiffPDE-XS, SimDiffPDE-S, SimDiffPDE-B, SimDiffPDE-L and SimDIffPDE-XL, to balance the inference speed and performance for various deployment requirements. 3) For flexiblity, we demonstrate our SimDiffPDE can adapt well to different input resolutions with minor modifications. 4) Lastly, our SimDiffPDE showcases the good feasibility to various PDE equations, including Navier-Stokes equation, Darcy flow equation, hyper-elastic problem and plastic forging problem.

In summary, the contributions of this paper can be outlined as follows: (1) We propose a simple yet high-performing generative baseline model for solving various PDEs, named SimDiffPDE. This model achieves consistent leading results across six datasets covering various grid types and PDE types, improving performance by an average of 22.0% compared to the second-best model, without complex network architectures or tailored designs. (2) We leverage a multi-scale noise strategy that further unlocks the potential of diffusion models in solving PDEs, enabling efficient capture of information at different frequencies and precise construction of the solution distribution for PDEs.

## 2    RELATED WORK

### 2.1    DIFFUSION MODEL

Diffusion models have been widely applied to various tasks, including image generation (Ho et al., 2022a), image restoration (Xia et al., 2023), super-resolution (Li et al., 2022a), text-to-image generation (Ruiz et al., 2023), video generation (Ho et al., 2022b), and audio generation (Liu et al., 2023). Additionally, diffusion models have been used to generate datasets related to PDEs (Lienen et al., 2023). Among these, Denoising Diffusion Probabilistic Models (DDPM) (Ho et al., 2020) are widely utilized. This model achieves data generation through a forward noise-adding process and a reverse denoising process. In the forward process, noise is gradually added to the real data until it approximates a standard normal distribution. In the reverse process, the model learns the conditional probability distribution between input conditions and output results, gradually denoising from pure noise to recover a high-quality target distribution. Leveraging the ability of DDPM to learn the probability distribution between input conditions and output results, we apply it to solve PDEs.

### 2.2    DEEP LEARNING PDEs SOLVER

For a long time, various numerical methods (Smith, 1985; Moukalled et al., 2016) have been widely used to solve PDEs. With the rise of deep learning, physics-informed neural networks (PINNs) (Raissi et al., 2019); the other class is data-driven neural operators. **Physics-informed neural networks** was proposed by  Raissi et al. (2019), where the constraints of PDEs (including equations, boundary conditions, and initial conditions) are used as a loss function. By employing a self-supervised learning approach to train neural networks (Ren et al., 2022; Yu et al., 2022), the model's output gradually conforms to these constraints, resulting in an approximate solution. However, this paradigm requires a rigorous formalization of partial differential equations and relies heavily on network optimization, which limits its practicality. **Neural operators**   establishes the mapping between inputs and outputs through neural operators, widely applied in the solution of partial differential equations (PDEs) (Li et al., 2020). The core idea of this operator is to approximate integration using linear projections in the Fourier domain. Based on this foundation, many improvements have emerged. For instance, U-FNO (Wen et al., 2022) and U-NO (Rahman et al., 2022) have proposed using the U-Net (Ronneberger et al., 2015) architecture to enhance the performance of FNO. F-FNO (Tran et al., 2021) utilizes factorization in the Fourier domain, while WMT (Gupta et al., 2021) introduces a neural operator learning scheme based on multiwavelets.

With the rise of Transformers (Vaswani, 2017), the recently high-performing Transolver (Wu et al., 2024) on multiple PDE benchmarks propose to construct mappings of inputs to outputs by learning the intrinsic physical states of the PDEs captured by learnable slices. However, these methods are essentially *deterministic*, which is not robust due to the chaotic nature of some PDEs. In contrast, SimDiffPDE leverages the characteristics of diffusion models to establish complex probability distributions between input conditions and output results. Simultaneously, through a multi-scale noise approach, it explicitly distinguishes and learns mutliscale information in PDE solution space.

## 3    SIMDIFFPDE: SIMPLE DIFFUSION BASELINE FOR SOLVING PARTIAL DIFFERENTIAL EQUATIONS

### 3.1    PDE SOLVING AS DIFFUSION GENERATIVE FORMULATION

We approach solving partial differential equations (PDEs) as a conditional denoising diffusion generation task. Specifically, we define PDEs over an input domain $\Omega \subset \mathbb{R}^{C_{\mathbf{x_g}}}$, where $C_{\mathbf{x_g}}$ denotes the

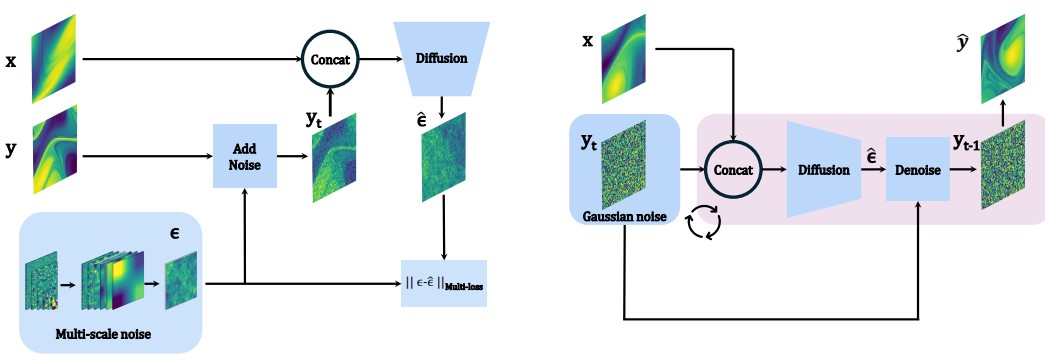

**Figure 2:** *Left*: Structure diagram of the SimDiffPDE training phase. *Right*: Structure diagram of the SimDiffPDE inference phase.

dimension of input space, and then discretize $\Omega$ into $N$ mesh points, represented as $\mathbf{x_g} \in \mathbb{R}^{N \times C_{\mathbf{x_g}}}$. Our goal is to train SimDiffPDE to model the conditional distributions $\mathbf{y} = D(\mathbf{y}|\mathbf{x})$ as the solution of PDE, where $\mathbf{x}$ combines geometric inputs $\mathbf{x_g}$ and observed quantities $\mathbf{x_u} \in \mathbb{R}^{N \times C_{\mathbf{x_u}}}$. Therefore, the complete input is $\mathbf{x} \in \mathbb{R}^{N \times C_{\mathbf{x}}}$, with $C_{\mathbf{x}} = C_{\mathbf{x_g}} + C_{\mathbf{x_u}}$.

In the forward process, starting from the conditional distribution at $\mathbf{y_0} := \mathbf{y}$, Gaussian noise is gradually added over time steps $t \in \{1, 2, 3, \cdots, T\}$ to obtain the noisy samples: $\mathbf{y_t}$ as

$$\mathbf{y_t} = \sqrt{\bar{\alpha}_t}\mathbf{y_0} + \sqrt{1 - \bar{\alpha}_t}\boldsymbol{\epsilon}, \tag{1}$$

where $\boldsymbol{\epsilon} \sim \mathcal{N}(0, \mathbf{I})$, $\bar{\alpha}_t := \prod_{s=1}^{t} 1 - \beta_s$, and $\{\beta_1, \beta_2, \beta_3, \cdots, \beta_T\}$ represents the variance schedule of a process over T steps. In the reverse process, the conditional denoising model $\boldsymbol{\epsilon}_\theta(\cdot)$, which is parameterized by learned parameters $\theta$, progressively removes noise from $\mathbf{y_t}$ to obtain $\mathbf{y_{t-1}}$.

During training, parameters $\theta$ are updated by taking a data pair $(\mathbf{x}, \mathbf{y})$ from the training data. At a random time step $t$, noise $\boldsymbol{\epsilon}$ is applied to $\mathbf{y}$, and the noise estimate $\hat{\boldsymbol{\epsilon}} = \boldsymbol{\epsilon}_\theta(\mathbf{y_t}, \mathbf{x}, t)$ is calculated. One of the denoising objective function is minimized, with a noise objective $\mathcal{L}$ as follows:

$$\mathcal{L}_{Multi} = \mathbb{E}_{\mathbf{y_0}, \boldsymbol{\epsilon} \sim \mathcal{N}(0, \mathbf{I}), t \sim U(T)} \|\boldsymbol{\epsilon} - \hat{\boldsymbol{\epsilon}}\|_{Multi} = \mathbb{E}_{\mathbf{y_0}, \boldsymbol{\epsilon} \sim \mathcal{N}(0, \mathbf{I}), t \sim U(T)} (\|\boldsymbol{\epsilon} - \hat{\boldsymbol{\epsilon}}\|_1 + \|\boldsymbol{\epsilon} - \hat{\boldsymbol{\epsilon}}\|_2), \tag{2}$$

where $\|\cdot\|_1$ and $\|\cdot\|_2$ denote $L_1$ and $L_2$ norm, respectively. During inference, $\mathbf{y} := \mathbf{y_0}$ is reconstructed from a normally distributed variable $\mathbf{y_t}$ by the learned denoiser $\boldsymbol{\epsilon}_\theta(\mathbf{y_t}, \mathbf{x}, t)$ iteratively.

## 3.2 NETWORK ARCHITECTURE

**Architecture** We propose a simple yet highly effective baseline model for PDEs based on diffusion models, while exploring their potential in this context. To achieve this, we keep the architecture straightforward, avoiding complex modules and elaborate tricks, even though these could potentially enhance the model's performance. To ensure the simplicity of the baseline model, we adopt the standard diffusion transformer block with AdaLN-Zero from Peebles & Xie (2023). The overall framework of SimDiffPDE is shown in Figure 2.

**Training phase** During training phase, we randomly select the input $\mathbf{x}$ and its corresponding output $\mathbf{y}$ from the training set of the PDEs, and then add multi-scale noise $\boldsymbol{\epsilon}_{Multi} \in \mathbb{R}^{N \times C_{\mathbf{y}}}$ (described in Sec. 3.3) to $\mathbf{y}$ to obtain noisy $\mathbf{y_t}$. Next, we concatenate the noisy $\mathbf{y_t} \in \mathbb{R}^{N \times C_{\mathbf{y}}}$ and $\mathbf{x} \in \mathbb{R}^{N \times C_{\mathbf{x}}}$ along the feature dimension to obtain $\mathbf{s} \in \mathbb{R}^{N \times C_{\mathbf{s}}}$, where $C_{\mathbf{s}} = C_{\mathbf{x}} + C_{\mathbf{y}}$. Then, we input $\mathbf{s}$ into the diffusion transformer block. When inputting $\mathbf{s}$ into the diffusion with transformer, the first step is to perform patch embedding on $\mathbf{s}$ and time embedding on time step $t \in \mathbb{R}^{N \times C_{\mathbf{s}}}$ used for the diffusion process. Finally, we input the embedded variables into the diffusion transformer block to predict noise $\hat{\boldsymbol{\epsilon}} \in \mathbb{R}^{N \times C_{\mathbf{y}}}$. In the training process, we use the loss function mentioned in Eq. 2. Experiments show that adding the $L_1$ loss on top of the $L_2$ loss can more effectively capture high-frequency information in the solution domain of PDEs.

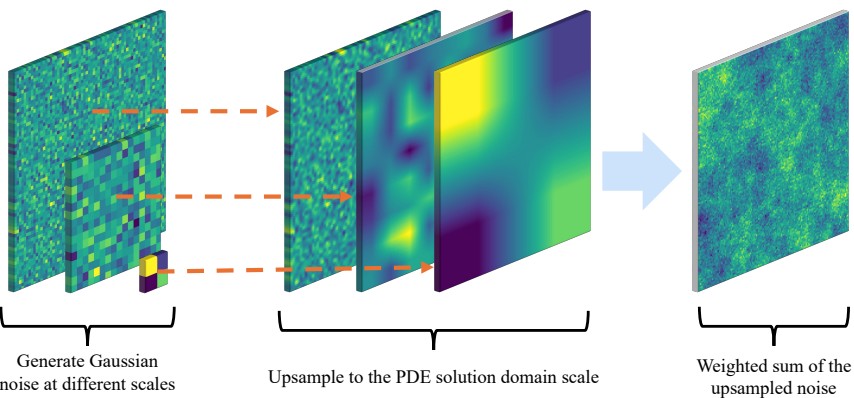

Figure 3: Visualization of the multi-scale noise implementation process. First, generate standard Gaussian noise of varying sizes, then upsample this noise to match the dimensions of the PDE solution domain, and finally, linearly combine the upsampled noise to create multi-scale noise.

**Inference phase** In the inference process of SimDiffPDE, it start with sampling from a standard Gaussian distribution $\mathbf{y_t} \in \mathbb{R}^{N \times C_{\mathbf{y}}}$. Next, we concantenate the $\mathbf{y_t}$ and the input conditions $\mathbf{x}$ of the PDEs along the feature dimension and fed into the trained diffusion transformer block. During the execution of the time steps, SimDiffPDE gradually denoise to ultimately generate the solution $\hat{\mathbf{y}} \in \mathbb{R}^{N \times C_{\mathbf{y}}}$ corresponding to the PDE. We further leverage the test-time ensemble for better solutions, which will be described in Sec. 3.4.

### 3.3 MULTI-SCALE NOISE

We propose a multi-scale noise approach to enhance the diffusion model's ability to capture and effectively relate various frequency noises. Specifically, as shown in Figure 3, our process has the following steps. First, Given the resolution $n \times n$ of the PDE's resolution domain, we generate the Gaussian noise $\epsilon_k \sim \mathcal{N}(0, \mathbf{I})$ with a resolution of $m_k \times m_k$, where $m_k \leq n$. Second, we upsample the different scales of Gaussian noise $\epsilon_k$ generated in Step 1 to match the size of the PDE solution domain, resulting in the noise $\epsilon'_k$ with the resolution of $n \times n$ through linear interpolation. Finally, we obtain the final noise $\epsilon_{Multi}$ through a weighted linear combination $\epsilon_{Multi} = \sum_{k=0}^{K} w_k \epsilon'_k$, where $\epsilon_{Multi}$ with the resolution of $n \times n$. The implementation of this approach is illustrated in Algorithm 1 in Appendix C.1. In the follows, we discuss how multi-scale Guassian noise improves PDE solver.

*Remark 3.3.1* (**Using Guassian noise is less efficient to destroy low-frequency flow pattern than using multi-scale noise in** *forward* **process.**) The default Guassian noise can not efficiently destroy the low-frequency pattern because default implementation samples every pixel from Guassian distribution independently and therefore its frequency is rather high. However, the proposed multi-scale noise ensembles noises with various frequencies, which shows better abilities to destroy patterns of various frequency. Empirically, we show noisy inputs which add 100 single-scale and multi-scale Guassian noise in the forward process (Figure 4). It is evident that multi-scale Guassian noise is more efficient to destroy the low-frequency pattern of solution domain. We claim the observation also applies to other noisy steps and illustrate the solution map added 1, 10, 50 and 500 steps of noise in Appendix D.2. We find that, as shown in Table 9, using multi-scale noise can significantly improve the accuracy of solving low-frequency information within the solution domain of PDEs. We can more intuitively illustrate this improvement using Figure 7 (*Bottom Right*).

*Remark 3.3.2* (**Using multi-scale noise can more effectively capture patterns of large scales, i.e., low-frequency information**). The core of diffusion models is to destroy the pattern and map them to Guassian distribution in the *forward* process and require the model to reconstruct the pattern by deep learning models in the *backward* process. The single-scale Guassian noise can not effectively destroy the low-frequency information, which leads the diffusion model inefficiently learning low-frequency information and large-scale patterns. In contrast, multi-scale noise can more effectively

**Ground-truth solution map**

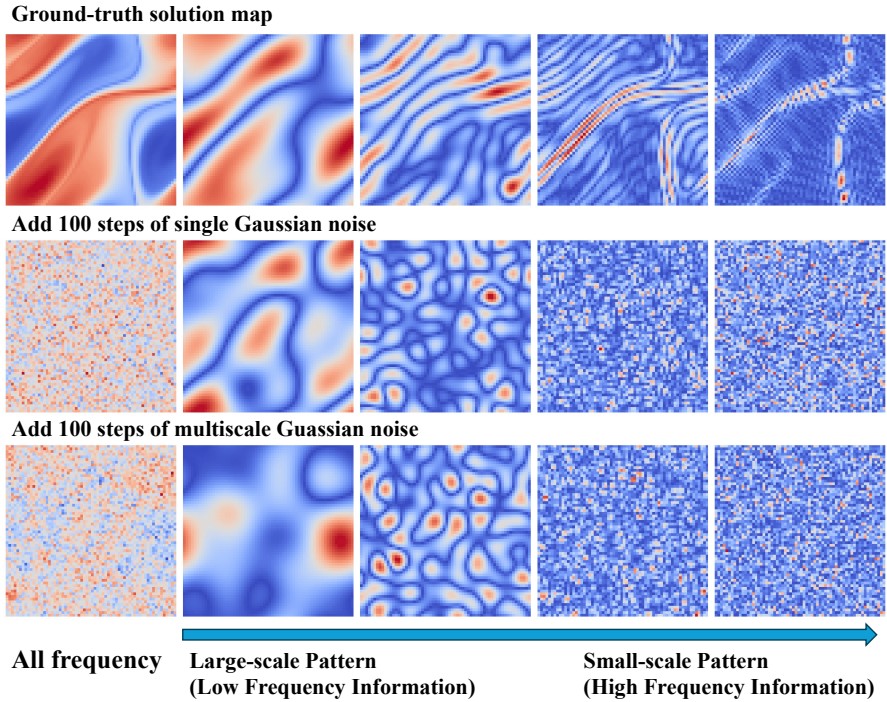

**Add 100 steps of single Gaussian noise**

**Add 100 steps of multiscale Guassian noise**

**All frequency**   **Large-scale Pattern**
                    **(Low Frequency Information)**

**Small-scale Pattern**
**(High Frequency Information)**

Figure 4: Illustration of the noisy solution maps at different frequencies using Guassian noise and multi-scale noise, respectively. Guassian and multi-noise perturbations are applied to the original images 100 times each, followed by Fourier and inverse Fourier transforms to extract different frequency components(0-3, 3-7, 7-20, 20-56) based on their distance from the zero-frequency point.

destroy both large-scale and small-scale patterns in the *forward* process, which can enforce the diffusion model to learn and reconstruct especially low-frequency information in solution domain.

### 3.4 TEST-TIME ENSEMBLE

Due to the nonlinear nature of PDEs, the small variation of the input parameters or states can lead to significant variations of solutions in some PDEs. With the stochastic nature of the DDPM inference process, different initial noises $\mathbf{y_t}$ can lead to varying solutions, which allows SimDiffPDE to simulate the nonlinear dynamics of PDEs. To better leverage this feature, we leverage a testing-time ensemble strategy for more accurate and robust solutions of PDEs.

Given the same input $\mathbf{x}$, we obtain a series of solutions $\{\mathbf{y}_1, \mathbf{y}_2, \cdots, \mathbf{y}_n\}$. We employ an iterative method to estimate the scale factors $\hat{s}_i$ and translations $\hat{t}_i$ of these solutions relative to a specific range. Due to the continuity and smoothness of PDE solutions, we achieve alignment of the solutions by minimizing the distance between pairs of transformed solutions $(\hat{\mathbf{y}}'_i, \hat{\mathbf{y}}'_j)$. Specifically, $\hat{\mathbf{y}}' = \hat{\mathbf{y}} \times \hat{s} + \hat{t}$. In each optimization step, we compute the median of the single solution points in the PDE solution domain as $\mathbf{m}(\mathbf{a}, \mathbf{b}) = \text{median}(\hat{\mathbf{y}'}_\mathbf{1}(\mathbf{a}, \mathbf{b}), \hat{\mathbf{y}'}_\mathbf{2}(\mathbf{a}, \mathbf{b}), \cdots, \hat{\mathbf{y}'}_\mathbf{n}(\mathbf{a}, \mathbf{b}))$ to derive the merged PDE solution. To prevent the solutions from converging to a trivial solution (e.g., all solutions being the same) and to ensure that $\mathbf{m}$ maintains an intensity within the unit range, we introduce an additional regularization term $\mathcal{R} = |\min(\mathbf{m})| + |1 - \max(\mathbf{m})|$. Therefore, the objective function can be expressed as

$$\min_{\substack{s_1, s_2, \cdots, s_n \\ t_1, t_2, \cdots, t_n}} \left( \sqrt{\frac{1}{B_n} \sum_{i=1}^{n-1} \sum_{j=i+1}^{n} \|\hat{\mathbf{y}}'_i - \hat{\mathbf{y}}'_i\|_2^2} + \lambda \mathcal{R} \right), \tag{3}$$

Table 1: Summary of experimental benchmarks, including different types of PDEs. Mesh denotes the size of the discrete mesh.

| Geometry | Benchmarks | Dimension | Mesh |
|---|---|---|---|
| Point Cloud | Elasticity | 2D | 972 |
| Structured Mesh | Plasticity | 2D+TIME | 3,131 |
| | Airfoil | 2D | 11,271 |
| | Pipe | 2D | 16,641 |
| Regular Grid | Navier-Stokes | 2D+TIME | 4,096 |
| | Darcy | 2D | 7,225 |

Table 2: Performance comparison based on six benchmarks, showing relative $L_2$ error ($\downarrow$). Lower values indicate better performance. "/" indicates that the baseline is not applicable to this benchmark. Relative promotion refers to the relative error reduction with respect to the second best model: Relative Promotion $= 1 - \frac{\text{Our error}}{\text{Second best error}}$ on each benchmark.

| Model | Point Cloud | Structured Meshes | | | Regular Grids | |
|---|---|---|---|---|---|---|
| | Elasticity | Plasticity | Airfoil | Pipe | Navier-Stokes | Darcy |
| FNO (Li et al., 2020) | / | / | / | / | 0.1556 | 0.0108 |
| WMT (Gupta et al., 2021) | 0.0359 | 0.0076 | 0.0075 | 0.0077 | 0.1541 | 0.0082 |
| U-FNO (Wen et al., 2022) | 0.0239 | 0.0039 | 0.0269 | 0.0056 | 0.2231 | 0.0183 |
| geo-FNO (Li et al., 2023) | 0.0229 | 0.0074 | 0.0138 | 0.0067 | 0.1556 | 0.0108 |
| U-NO (Rahman et al., 2022) | 0.0258 | 0.0034 | 0.0078 | 0.0100 | 0.1713 | 0.0113 |
| F-FNO (Tran et al., 2021) | 0.0263 | 0.0047 | 0.0078 | 0.0070 | 0.2322 | 0.0077 |
| LSM (Wu et al., 2022) | 0.0218 | 0.0025 | 0.0059 | 0.0050 | 0.1535 | 0.0065 |
| Galerkin (Cao, 2021) | 0.0240 | 0.0120 | 0.0118 | 0.0098 | 0.1401 | 0.0084 |
| HT-Net (Liu et al., 2022) | / | 0.0333 | 0.0065 | 0.0059 | 0.1847 | 0.0079 |
| Oformer (Li et al., 2022b) | 0.0183 | 0.0017 | 0.0183 | 0.0168 | 0.1705 | 0.0124 |
| GNOT (Hao et al., 2023) | 0.0086 | 0.0336 | 0.0076 | 0.0047 | 0.1380 | 0.0105 |
| FactFormer (Li et al., 2024a) | / | 0.0312 | 0.0071 | 0.0060 | 0.1214 | 0.0109 |
| ONO (Xiao et al., 2023) | 0.0118 | 0.0048 | 0.0061 | 0.0052 | 0.1195 | 0.0076 |
| Transolver (Wu et al., 2024) | 0.0064 | 0.0012 | 0.0053 | 0.0033 | 0.0900 | 0.0057 |
| SimDiffPDE-S | 0.0057 | 0.0010 | 0.0049 | 0.0030 | 0.0529 | 0.0050 |
| Relative Promotion | 10.9% | 16.7% | 7.5% | 9.1% | 41.2% | 12.2% |
| SimDiffPDE-B | 0.0049 | 0.0009 | 0.0047 | 0.0028 | 0.0437 | 0.0045 |
| Relative Promotion | 23.4% | 25.0% | 11.3% | 15.2% | 51.4% | 21.1% |
| SimDiffPDE-L | 0.0043 | 0.0008 | 0.0043 | 0.0024 | 0.0394 | 0.0041 |
| Relative Promotion | 32.8% | 33.3% | 18.9% | 27.3% | 56.2% | 28.1% |
| SimDiffPDE-XL | 0.0039 | 0.0007 | 0.0040 | 0.0022 | 0.0355 | 0.0037 |
| Relative Promotion | 39.1% | 41.7% | 24.5% | 33.3% | 60.6% | 35.1% |

where the binomial coefficient $B_n = \binom{n}{2}$ indicates the total number of possible combinations of solution pairs from $n$ solutions. After iterative optimization, the merged solution $\mathbf{m}$ is regarded as our ensemble solution. Due to the diversity and complexity of PDE solutions, this integration step does not require ground truths. Through multiple inferences, we can capture the spatial features and dynamic variations of the solutions, thereby enhancing the robustness and accuracy of predictions.

## 4 EXPERIMENT

**Benchmarks**  Our experiments cover various types of PDEs, including point clouds, structured meshes, and regular grids, as shown in Table 1. The Navier-Stokes and Darcy equations were introduced by Li et al. (2020), while Elasticity, Plasticity, and Airfoil problems were proposed by Li et al. (2023), all of which are widely followed.

**Baselines**  We comprehensively compare SimDiffPDE with baseline , including neural operators like FNO (Li et al., 2020), Transformer-based solvers such as GNOT (Hao et al., 2023), and the recent state-of-the-art Transolver (Wu et al., 2024).

Table 3: Compared to the series of benchmarks proposed by Li et al. (2020), the performance of the model at different Reynolds numbers, showing the relative $L_2$ error ($\downarrow$). The smaller, the better. Where $\nu$ represents viscosity, $T$ is the discrete time step, and $N$ is the size of the training dataset.

| Model | $\nu = 1e - 3$ $T = 50$ $N = 1000$ | $\nu = 1e - 4$ $T = 30$ $N = 1000$ | $\nu = 1e - 5$ $T = 20$ $N = 1000$ |
|---|---|---|---|
| FNO-3D | 0.0086 | 0.1918 | 0.1893 |
| FNO-2D | 0.0128 | 0.1559 | 0.1556 |
| U-Net | 0.0245 | 0.2051 | 0.1982 |
| TF-Net | 0.0225 | 0.2253 | 0.2268 |
| Res-Net | 0.0701 | 0.2871 | 0.2753 |
| **SimDiffPDE-B** | **0.0062** | **0.0342** | **0.0437** |

## 4.1 MAIN RESULTS

To clearly benchmark our model among various PDE solvers, we first conduct experiments on six well-established datasets, which can be easily obtained from previous studies (Hao et al., 2023; Wu et al., 2024) to create a comprehensive leaderboard.

**Point clouds**   For point cloud-based tasks, SimDiffPDE achieves a significant improvement over competing methods. Specifically, in the elasticity task, SimDiffPDE outperforms the previous best model, Transolver (Wu et al., 2024), by a margin of 23.4%, with an impressive relative $L_2$ error of 0.0049. This demonstrates the model's strong ability to handle irregular and unstructured data, making it highly effective for point cloud applications.

**Structured meshes**   SimDiffPDE also excels in tasks utilizing structured meshes, which are frequently employed in simulations of plasticity, airfoil flow, and pipe flow. In the plasticity task, SimDiffPDE achieves a relative error of just 0.0009, representing a 25.0% improvement over the next best model. Similarly, it achieves relative improvements of 11.3% and 15.2% in the airfoil and pipe tasks, respectively, further demonstrating its superiority in structured mesh-based problems.

**Regular grids**   In the most demanding benchmarks, based on regular grids, SimDiffPDE sets a new benchmark, particularly in the Navier-Stokes and Darcy flow tasks. In the Navier-Stokes benchmark, SimDiffPDE shows a remarkable 51.4% improvement with a relative $L_2$ error of 0.0437, far outperforming the nearest competitor. Table 3 demonstrates that SimDiffPDE achieves leading accuracy in solving the Navier-Stokes equations at different Reynolds numbers, indicating that our model can effectively apply to the Navier-Stokes equations across various Reynolds numbers, highlighting the feasibility of SimDiffPDE. In the Darcy benchmark, the model delivers a 21.1% improvement, further solidifying its effectiveness. These results underline SimDiffPDE's ability to accurately capture complex dynamics in regular grid simulations. Furthermore, as shown in Table 11 in Appendix, SimDiffPDE outperforms the second-best model, Transolver (Wu et al., 2024), in solving the Darcy benchmark across different resolutions.

## 4.2 ABLATION STUDY

**Training noise**   To verify whether the use of multi-scale noise can better capture the frequency information in the solution domain of PDEs, especially low-frequency information, we conducte a comparative experiment. As shown in Table 4, we analyze the errors in the model's generated results for high-frequency and low-frequency components under two training noise conditions. The experimental results indicate that using multi-scale noise improves the model's precision in generating both low-frequency and high-frequency information, with a particularly significant enhancement in low-frequency generation. Table 9 in Appendix C.1 presents the full-frequency errors of five benchmarks under two types of training noise. It was found that training with multi-scale noise reduced the average error across the five benchmarks by 12.4% compared to using Gaussian noise. This result further demonstrates the effectiveness of multi-scale noise. Appendix B.3 shows the details of the ablation experiment implementation.

Table 4: Comparison of Gaussian noise and multi-scale noise on training results, showing the relative $L_2$ error ($\downarrow$). Relative promotion refers to the relative error reduction with respect to the second best model: Relative Promotion $= 1 - \frac{\text{Our error}}{\text{Second best error}}$ on each benchmark. GN - Gaussian Noise, MN - Multi-scale Noise.

| Benchmark | High Frequency Error | | | Low Frequency Error | | |
|---|---|---|---|---|---|---|
| | GN | MN | Relative Promotion | GN | MN | Relative Promotion |
| Plasticity | 0.0285 | 0.0253 | 11.2% | 0.0010 | 0.0008 | 20.0% |
| Airfoil | 0.1153 | 0.1038 | 10.0% | 0.0051 | 0.0042 | 17.6% |
| Pipe | 0.0964 | 0.0872 | 9.5% | 0.0029 | 0.0024 | 17.2% |
| Navier–Stokes | 0.2062 | 0.1857 | 9.9% | 0.0499 | 0.0406 | 18.6% |
| Darcy | 0.1076 | 0.0954 | 11.3% | 0.0048 | 0.0037 | 22.9% |

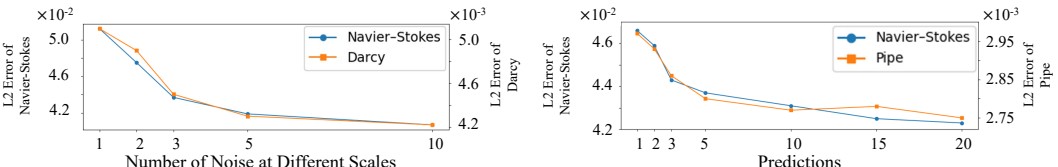

Figure 5: *Left*: Comparison of the impact of different quantities of noise at various scales on solution error. *Right*: Comparison of the impact of different ensemble sizes on solution error.

To further investigate the validity of multi-scale noise, we choose two representative benchmarks with more high-frequency and low-frequency components, respectively: Navier-Stokes, Darcy. By adjusting the number of noise components in the multi-scale noise, we find that, as shown in Figure 5 (*Left*), training with noise composed of three different scales can reduce the error by 13.2%, and training with noise composed of five different scales leads to an 16.9% improvement on average. It was observed that, due to the limited size of the PDE solution domain, marginal improvements gradually decrease when the number of different scales exceeds five. It should be noted that using more noise at different scales implies that the selected noise will have smaller scale differences. Appendix C.1 provides further details.

**Test-time ensembling** We conduct tests to evaluate the effectiveness of the proposed test-time ensembling scheme by aggregating various quantities of predictions in the benchmarks of Navier–Stokes and Pipe. As shown in Figure 5 (*Right*), a single prediction from SimDiffPDE yields quite good results. Ensemble of 5 predictions reduces the relative error on Navier–Stokes by about 5.0%, while ensemble of 10 predictions provides an improvement of approximately 7.5%. It is observed that, as a system effect, performance steadily increases with the number of predictions, but the marginal improvements decrease when the number of predictions exceeds 10.

### 4.3 MODEL ANALYSIS

**Scalability of data size and model size** Our proposed SimDiffPDE show good scalbility on both data and model size. As shown in Figure 6 (*Right*), we select different number of training samples from Darcy, and verify that the SimDiffPDE-B can consistently achieve lower errors with the increasing number of training samples. We also verify that our propose SimDiffPDE have good scalability on model size in Table 2. We demonstrate that the error of PDE solution consistently decreases with the model size increasing from SimDiffPDE-S to SimDiffPDE-XL. These findings provide a solid foundation for the application of large-scale PDE solvers.

**Flexibility to various resolutions** To validate the flexibility of SimDiffPDE, we tested inputs at different resolutions, with disparities reaching up to 100 times. The results indicate that SimDiffPDE performs consistently well across all resolutions, as shown in Figure 6 (*Left*), with its solution accuracy consistently surpassing that of the second-best model. Table 11 Appendix C.3 provides specific numerical comparisons. These results demonstrate the robust flexibility of SimDiffPDE.

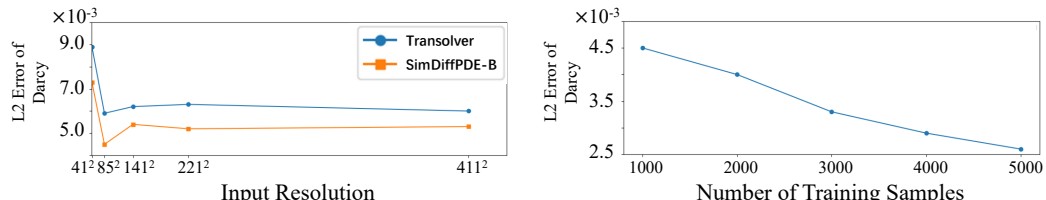

Figure 6: *Left*: Comparison of the solving performance of Transolver (Wu et al., 2024) and SimDiffPDE on Darcy benchmarks at different resolutions. *Right*: Comparison of the effects of different training sample sizes on solution accuracy.

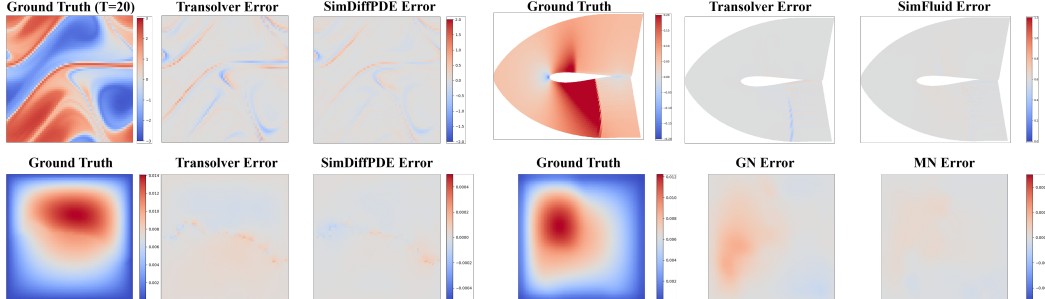

Figure 7: Case study on error maps. *Top Left* image shows the performance of Transolver (Wu et al., 2024) and SimDiffPDE on the Navier-Stokes benchmark, where SimDiffPDE significantly improves in regions with abundant high-frequency information, such as sharp boundaries. *Top Right* image compares the two methods on the Airfoil benchmark, highlighting that SimDiffPDE outperforms Transolver (Wu et al., 2024). *Bottom Left* image illustrates the performance of both methods on the Darcy benchmark, where SimDiffPDE excels in areas rich in low-frequency information, particularly in the center. *Bottom Right* image examines the impact of training with Gaussian noise versus multi-scale noise on solving low-frequency information in the Darcy benchmark, demonstrating that training with multi-scale noise significantly enhances the accuracy of low-frequency information solutions. GN - Gaussian Noise, MN - Multi-scale Noise.

**Case study**    To provide a clearer demonstration of the advantages of SimDiffPDE in solving different PDEs, we plot the error maps of various benchmarks, as shown in Figure 7. Compared to the second-best model, Transolver (Wu et al., 2024), SimDiffPDE exhibits significant improvements in the low-frequency region, such as the central area of the Darcy benchmark solution domain, shown as Figure 7 (*Bottom Left*). Additionally, in the high-frequency region, particularly at sharp boundaries within the Navier–Stokes benchmark solution domain, SimDiffPDE also achieves commendable advancements, shown as Figure 7 (*Top Left*). These results further confirm that SimDiffPDE effectively captures the features of different frequencies within the PDE solution domain and establishes an accurate solution distribution.

## 5    CONCLUSION AND FUTURE WORK

In this paper, we introduce SimDiffPDE, the first PDE solver based on a diffusion model with Transformers. Unlike traditional deep learning PDE solvers that create a deterministic mapping between input conditions and output results, SimDiffPDE employs DDPM and multi-scale noise to capture complex physical and geometric states across various frequencies in the PDE solution domain. This approach establishes a complex probability distribution between inputs and outputs, resulting in high-precision solutions. SimDiffPDE has achieved state-of-the-art performance on six widely recognized benchmarks. In the future, our goal is to extend SimDiffPDE to solve non-stationary PDEs in continuous time, similar to video generation, while also exploring large-scale pre-training of SimDiffPDE.

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

# A OVERVIEW

In this appendix, we provide detailed content that complements the main paper. Section B elaborates on the implementation details of the experiments, including benchmarks, evaluation metrics and frequency analysis of information in the solution domain of PDEs. Section C provides a more detailed analysis of experiments related to multi-scale noise, training strategies, and model flexibility. Section D presents a visual representation of the details of the SimDiffPDE denoising process and provides a visual analysis of the noise addition process using multi-scale noise.

# B IMPLEMENTATION DETAILS

## B.1 BENCHMARKS

We validate the performance of our model on three benchmarks: the Navier-Stokes equations, the Darcy flow equations, and the airfoil problem using Euler's equations. For detailed information about the benchmarks, please refer to Table 5. Our tests involve the following two types of PDEs:

- **Solid material** (Dym et al., 1973): Elasticity and Plasticity.
- **Navier-Stokes equations for fluid** (McLean, 2012): Navier-Stokes, Airfoil and Pipe.
- **Darcy's law** (Hubbert, 1956): Darcy.

The following are the detailed information for each benchmark.

**Elasticity** This benchmark evaluates the internal stress distribution within an elastic material based on its structural configuration, discretized into 972 points (Li et al., 2023). For each sample, the input is a tensor of shape $972 \times 2$, representing the 2D coordinates of the discretized points. The output is the corresponding stress at each point, resulting in a tensor of shape $972 \times 1$. The dataset consists of 1000 samples with varying structures for training, and an additional 200 samples are reserved for testing.

**Plasticity** This benchmark aims to predict the future deformation of a plastic material subjected to an impact from an arbitrarily shaped die (Li et al., 2023). Each input is a die shape, discretized into a structured mesh and stored as a tensor of shape $101 \times 31$. The output is the deformation at each mesh point over 20 time steps, represented by a tensor of shape $20 \times 101 \times 31 \times 4$, where the four channels capture the deformation in different directions. The dataset comprises 900 training samples with different die shapes, and 80 samples are used for testing.

**Navier-Stokes** This benchmark simulates incompressible viscous flow on a unit torus, where the fluid density is constant and the viscosity is set to $1e-3$, $1e-4$ and $1e-5$. The fluid field is discretized into a $64 \times 64$ regular grid. The task is to predict the future 10 steps of the fluid based on the observations from the previous 10 steps. The model is trained using 1,000 fluid instances with different initial conditions and tested with 200 new samples.

Table 5: The benchmarks Elasticity, Navier–Stokes, Darcy Flow, Plasticity, Pipe and Airfoil were created by Li et al. (2020). Dim represents the dimension of the dataset, Mesh refers to the size of the discretized grid, and Dataset includes the number of samples in the training and testing sets.

| Geometry | Benchmarks | Dim | Mesh | Input | Output | Dataset |
|---|---|---|---|---|---|---|
| Point Cloud | Elasticity | 2D | 972 | Structure | Inner Stress | (1000, 200) |
| Regular Grid | Navier–Stokes | 2D+Time | 4,096 | Past Velocity | Future Velocity | (1000, 200) |
| Regular Grid | Darcy Flow | 2D | 7,225 | Porous Medium | Fluid Pressure | (1000, 200) |
| Structured Mesh | Plasticity | 2D+Time | 3,131 | External Force | Mesh Displacement | (900, 80) |
| Structured Mesh | Airfoil | 2D | 11,271 | Structure | Mach Number | (1000, 200) |
| Structured Mesh | Pipe | 2D | 16,641 | Structure | Fluid Velocity | (1000, 200) |

s

**Airfoil**    This benchmark estimates the Mach number based on airfoil shapes. The input shapes are discretized into a structured grid of $221 \times 51$, and the output is the Mach number at each grid point (Li et al., 2023). All shapes are deformations of the NACA-0012 case provided by the National Advisory Committee for Aeronautics. A total of 1,000 different airfoil design samples are used for training, with an additional 200 samples for testing.

**Pipe**    This benchmark estimates the horizontal fluid velocity within a pipe based on its structural design (Li et al., 2023). The pipe is discretized into a structured mesh of size $129 \times 129$, resulting in an input tensor of shape $129 \times 129 \times 2$ that encodes the positions of the mesh points. The output is a velocity tensor of shape $129 \times 129 \times 1$, capturing the fluid velocity at each point. The dataset includes 1000 training samples with varying pipe geometries, and 200 test samples generated by modifying the pipe's centerline.

**Darcy**    This benchmark is utilized to simulate fluid flow through porous media (Li et al., 2020). In the experiment, the process is discretized into a regular grid of $421 \times 421$, and the data is downsampled to a resolution of $85 \times 85$ for the main experiments. The model's input is the structure of the porous medium, while the output is the fluid pressure at each grid point. A total of 1,000 samples are used for training and 200 samples for testing, covering various structures of the medium.

## B.2    METRICS

To visually demonstrate the state-of-the-art performance of our model and ensure fair comparison with other models, we choose to use relative $L_2$ to measure the error in the physics field. The relative $L_2$ error of the model prediction field $\hat{\phi}$ compared to the given physical field $\phi$ can be calculated as follows:

$$\text{Relative } L_2 \text{ Loss} = \frac{\|y - \hat{y}\|_2}{\|y\|_2} \tag{4}$$

## B.3    FREQUENCY ANALYSIS IN PDES

In this study, we introduce the use of high-frequency and low-frequency filters to analyze the differences between the generated solutions of PDEs and their ground truth solutions. These filters are

implemented through convolution operations to extract different frequency features from the images. The specific implementation is as follows:

### B.3.1 HIGH-FREQUENCY FILTER

**Definition**: The high-frequency filter is used to retain the high-frequency components of the image, primarily emphasizing edges and details. We choose a simple high-pass filter defined as follows:

```
highpass_kernel = np.array([[0, -1, 0],
                            [-1, 4, -1],
                            [0, -1, 0]], np.float32)
```

**Explanation**:

- This filter applies a large positive weight (4) to the center pixel and negative weights to the surrounding pixels, enhancing edge information.
- In the convolution operation, the filter subtracts the average value of surrounding pixels, highlighting areas with significant changes, thus achieving high-frequency component extraction.

### B.3.2 LOW-FREQUENCY FILTER

**Definition**: The low-frequency filter is used to smooth the image and remove high-frequency noise. We use a simple averaging filter defined as follows:

```
lowpass_kernel = np.ones((8, 8), np.float32) / 64
```

**Explanation**:

- This filter is an 8x8 averaging filter, where each element has a value of $1/64$ (the total sum of $8 \times 8$). This means that in the convolution operation, the filter calculates the average of the surrounding 64 pixels.
- By retaining low-frequency components, this filter effectively reduces high-frequency noise in the image, resulting in a smoother appearance.

By employing the aforementioned methods, we can effectively distinguish information of different frequencies within the solution domain of PDEs, enabling a series of related experiments.

## C  SUPPLEMENTARY ANALYSIS

### C.1  ANALYSIS OF EXPERIMENTS ON MULTI-SCALE NOISE

In the main text, we demonstrate that using multi-scale noise can better capture both high-frequency and low-frequency information in the solution domain of PDEs, leading to improved prediction results. Table 9 presents the generation effects of multi-scale noise across all frequencies in the PDEs solution domain.

Table 6: Comparison of the impact of different quantities of noise at various scales on solution error, showing the relative $L_2$ error ($\downarrow$).

| Number of Different Scales Noise | Navier–Stokes | Darcy |
|:---:|:---:|:---:|
| 1 | 0.0512 | 0.0051 |
| 2 | 0.0475 | 0.0049 |
| 3 | 0.0437 | 0.0045 |
| 5 | 0.0419 | 0.0043 |
| 10 | 0.0407 | 0.0042 |

In multi-scale noise, a scale pyramid is constructed by sampling multiple Gaussian noises. These Gaussian noises are then combined using upsampling, weighted averaging, and renormalization. The weight for the $i$-th layer of the pyramid is computed as $s_i$, where $0 < s < 1$ represents the intensity of the influence of different scales noise. To make this noise more akin to the Gaussian noise used in the original DDPM formulation, we suggest adjusting the weights of the layers $i > 0$ according to the diffusion schedule. Specifically, at time step $t$, the weight assigned to the $i$-th layer is given by $\left(\frac{st}{T}\right)^i$, where $T$ is the total number of diffusion steps. Moreover, as shown in Table 8, employing a cosine annealing strategy for sampling $s$ can further enhance the model's performance.

In addition, we conducted experiments on the number of Gaussian noises that make up the multi-scale noise, as shown in Table 6.

Table 7: Comparison of the impact of different of different loss strategies on solution error, showing the relative $L_2$ error ($\downarrow$).

| $L_1$ Loss | $L_2$ Loss | $L_2$ Error $\downarrow$ |
|---|---|---|
| ✓ | × | 0.3743 |
| × | ✓ | 0.0826 |
| ✓ | ✓ | **0.0437** |

Table 8: Comparison of the impact of different of different noise strategies on solution error, showing the relative $L_2$ error ($\downarrow$). GN - Gaussian Noise, AS - Annealing Strategy, MN - Multi-scale Noise.

| GN | AS | MN | $L_2$ Error $\downarrow$ |
|---|---|---|---|
| ✓ | × | × | 0.0732 |
| ✓ | ✓ | × | 0.0512 |
| × | × | ✓ | 0.0562 |
| × | ✓ | ✓ | **0.0437** |

Table 9: Comparison of solution accuracy using multi-scale noise and Gaussian noise, showing the relative $L_2$ error ($\downarrow$). Relative promotion refers to the reduction in error compared to training with Gaussian noise: Relative Promotion $= 1 - \frac{\text{Our error}}{\text{Second best error}}$ on each benchmark. GN - Gaussian Noise, MN - Multi-scale Noise.

| Benchmark | GN | MN | Relative Promotion |
|---|---|---|---|
| Plasticity | 0.0010 | 0.0009 | 10.0% |
| Airfoil | 0.0054 | 0.0047 | 13.0% |
| Pipe | 0.0032 | 0.0028 | 12.5% |
| Navier–Stokes | 0.0512 | 0.0437 | 14.6% |
| Darcy | 0.0051 | 0.0045 | 11.8% |

Moreover, to provide a more intuitive demonstration of our multi-scale noise construction process, we use pseudocode to better illustrate this procedure, as shown in Algorithm 1.

---

**Algorithm 1** Multi-scale Noise

**Input:** PDE's Solution $\mathbf{y}$, Number of Scales $k$, Strength $\alpha$, Upsampler $U$
$(b, c, w, h) \leftarrow \text{shape}(\mathbf{y})$      ▷ Get dimensions of PDE's solution
$\boldsymbol{\mathcal{E}}_{Multi} \leftarrow \text{randn}(b, c, w, h)$      ▷ Initialize Multi-scale noise
**for** $i = 0$ to $k - 1$ **do**      ▷ Loop over k iterations
  $r \leftarrow \text{rand}(1) \times 2 + 2$      ▷ Generate random scaling factor
  $w \leftarrow \max(1, \lfloor w/(r^i) \rfloor)$      ▷ Update width with scaling
  $h \leftarrow \max(1, \lfloor h/(r^i) \rfloor)$      ▷ Update height with scaling
  $\boldsymbol{\mathcal{E}}_{Multi} \leftarrow \boldsymbol{\mathcal{E}}_{Multi} + U(\text{randn}(b, c, w, h)) \times \alpha^i$      ▷ Add upsampled noise
  **if** $w == 1$ or $h == 1$ **then**      ▷ Check for minimum dimensions
   **break**      ▷ Exit loop if dimensions are 1
  **end if**
**end for**
**return** $\frac{\boldsymbol{\mathcal{E}}_{Multi}}{std(\boldsymbol{\mathcal{E}}_{Multi})}$      ▷ Return Multi-scale noise

---

## C.2  ANALYSIS OF EXPERIMENTAL STRATEGIES

As shown in Table 7, our experiments indicate that adding $L_1$ error guidance to $L_2$ error training improves training results. Additionally, Table 10 demonstrates that employing both multi-scale noise and multi-loss strategies significantly enhances model performance.

Table 10: Comparison of the impact of different of different training strategies on solution error, showing the relative $L_2$ error ($\downarrow$).

| Multi-scale  noise  +  Annealing | Multi-loss | $L_2$ Error $\downarrow$ |
|:---:|:---:|:---:|
| $\times$ | $\times$ | 0.2562 |
| $\checkmark$ | $\times$ | 0.2273 |
| $\times$ | $\checkmark$ | 0.0532 |
| $\checkmark$ | $\checkmark$ | **0.0437** |

## C.3  ANALYSIS OF FLEXIBILITY

As mentioned in the main text, our model exhibits strong flexibility, capable of handling inputs of varying resolutions while achieving state-of-the-art performance, as shown in Table11.

Table 11: Comparison of performance between SimDiffPDE and Transolver (Wu et al., 2024) across different mesh resolutions, showing the relative $L_2$ error ($\downarrow$). Relative promotion refers to the reduction in error compared to training with Gaussian noise: Relative Promotion $= 1 - \frac{\text{Our error}}{\text{Second best error}}$ on each benchmark.

| Number of Mesh Points (Resolution) | 1,681 (41×41) | 3,364 (58×58) | 7,225 (85×85) | 10,609 (103×103) | 19,881 (141×141) | 44,521 (211×211) | 168,921 (411×411) |
|---|---|---|---|---|---|---|---|
| Transolver (Wu et al., 2024) | 0.0089 | 0.0058 | 0.0057 | 0.0057 | 0.0062 | 0.0063 | 0.0060 |
| **SimDiffPDE** | **0.0073** | **0.0052** | **0.0045** | **0.0044** | **0.0054** | **0.0052** | **0.0053** |
| Relative Error Reduction | 18.0% | 10.3% | 21.1% | 22.8% | 12.9% | 17.5% | 11.7% |

# D  VISUALIZATION

## D.1  VISUALIZATION OF DENOISING PROCESS

To provide a clearer visualization of the inputs in the benchmark, the denoising process of SimDiff-PDE, and the comparison between the output results and the actual results, we have visualized this entire process. Please refer to the Figure 8 for details.

## D.2  VISUALIZATION OF ADDING NOISE USING MULTI-SCALE NOISE

In the main text, we present visualizations of adding noise to the original image using multi-scale noise and Gaussian noise over 100 time steps. To better illustrate this process, we will present visualizations of adding noise to the original image using multi-scale noise and Gaussian noise over 1, 10, 50, and 500 time steps. These correspond to Figure 9, Figure 10, Figure 11, and Figure 12, respectively. You can find these figures at the end of the appendix.

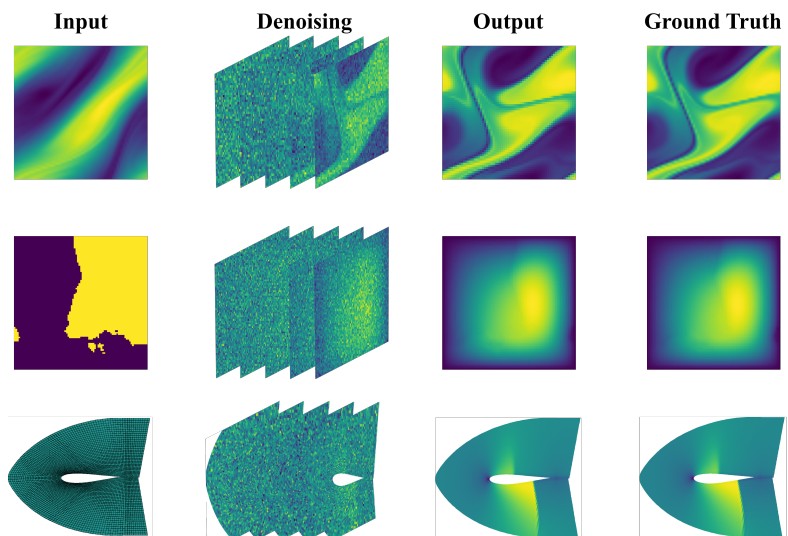

Figure 8: Denoising performance for PDEs: (1) Navier-Stokes Equation; (2) Darcy Flow Equation; (3) Airfoil Problem with Euler's Equation.

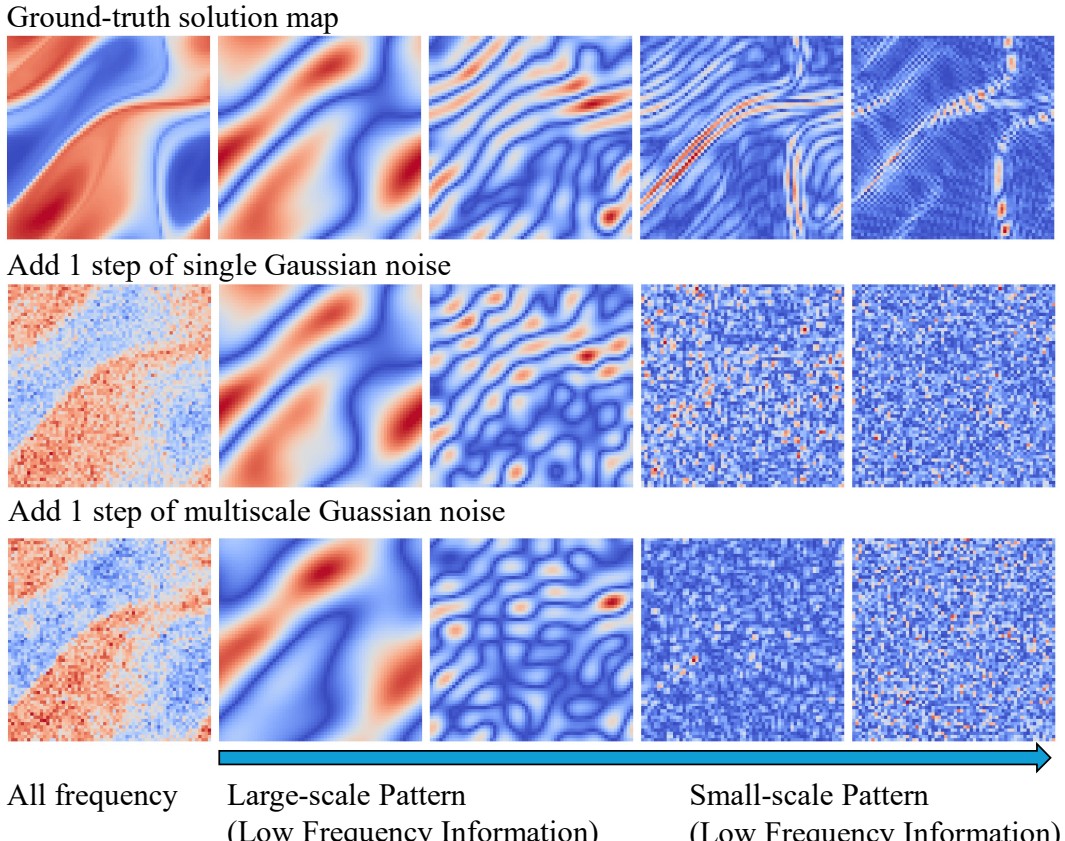

Figure 9: Illustration of the noisy solution maps at different frequencies using Guassian noise and multi-scale noise, respectively. Guassian and multi-noise perturbations are applied to the original images 1 time each, followed by Fourier and inverse Fourier transforms to extract different frequency components(0-3, 3-7, 7-20, 20-56) based on their distance from the zero-frequency point.

Ground-truth solution map

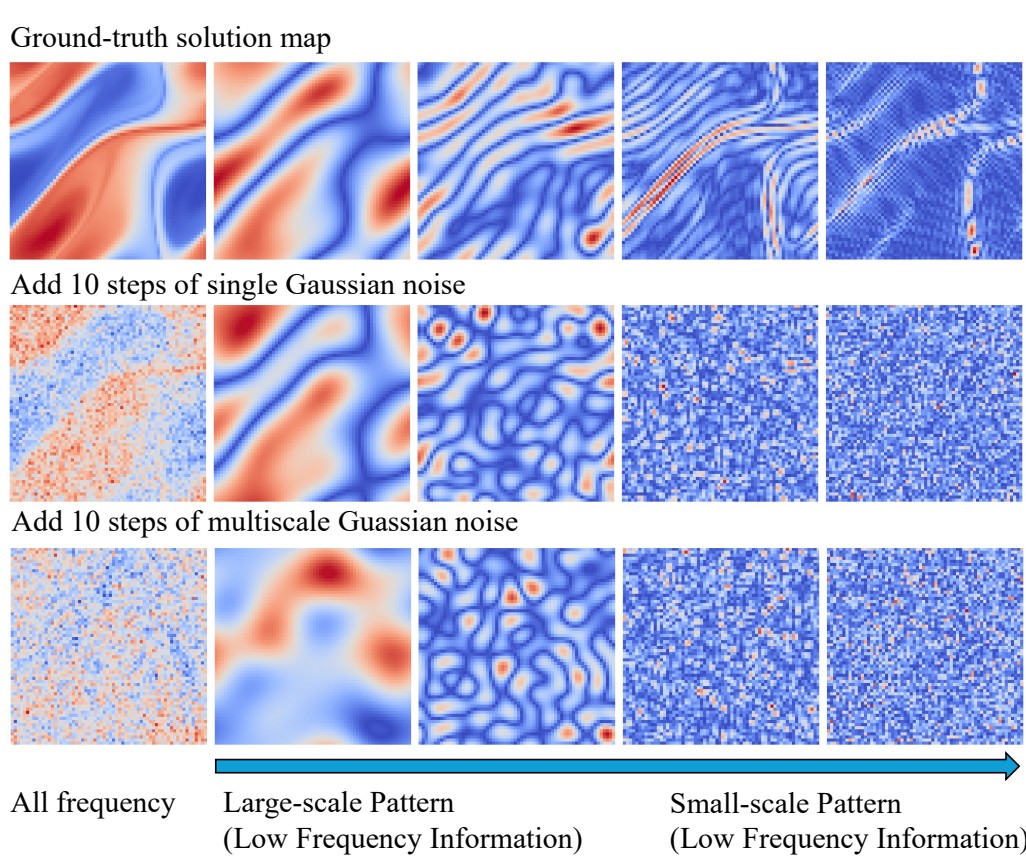

Add 10 steps of single Gaussian noise

Add 10 steps of multiscale Guassian noise

All frequency     Large-scale Pattern           Small-scale Pattern

(Low Frequency Information)     (Low Frequency Information)

Figure 10: Illustration of the noisy solution maps at different frequencies using Guassian noise and multi-scale noise, respectively. Guassian and multi-noise perturbations are applied to the original images 10 times each, followed by Fourier and inverse Fourier transforms to extract different frequency components(0-3, 3-7, 7-20, 20-56) based on their distance from the zero-frequency point.

Ground-truth solution map

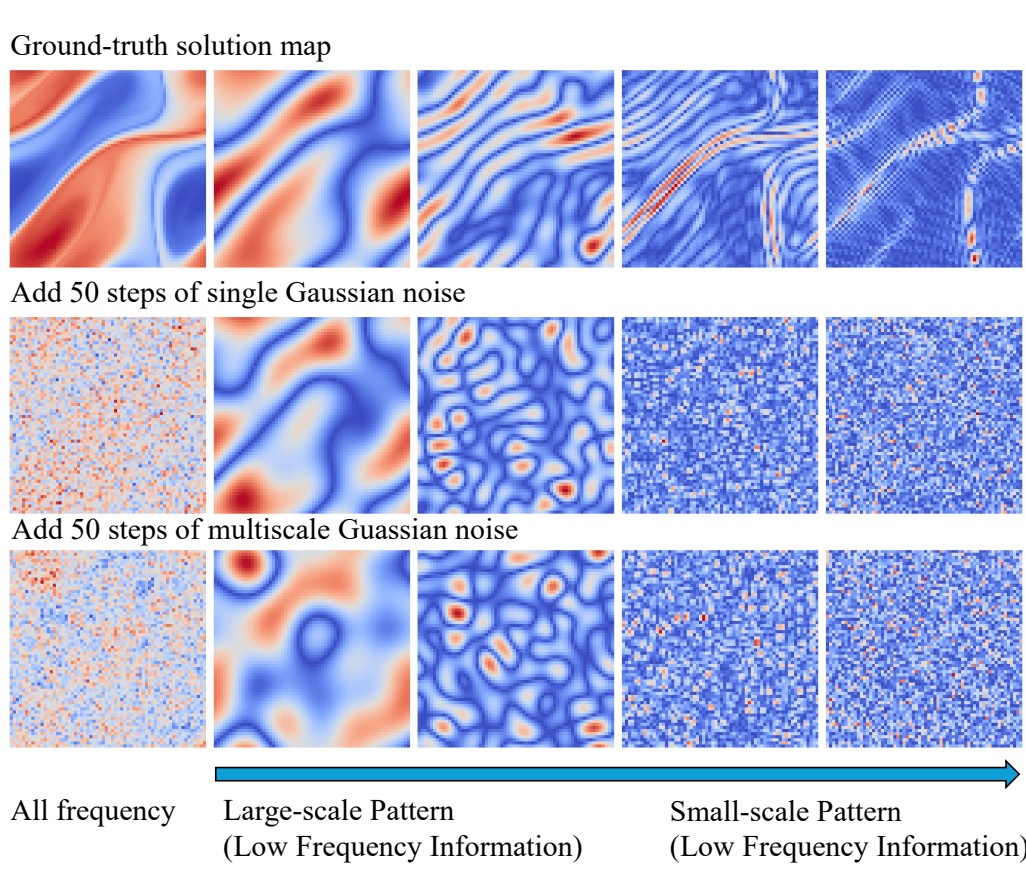

Add 50 steps of single Gaussian noise

Add 50 steps of multiscale Guassian noise

All frequency          Large-scale Pattern          Small-scale Pattern
                       (Low Frequency Information)   (Low Frequency Information)

Figure 11: Illustration of the noisy solution maps at different frequencies using Guassian noise and multi-scale noise, respectively. Guassian and multi-noise perturbations are applied to the original images 50 times each, followed by Fourier and inverse Fourier transforms to extract different frequency components(0-3, 3-7, 7-20, 20-56) based on their distance from the zero-frequency point.

Ground-truth solution map

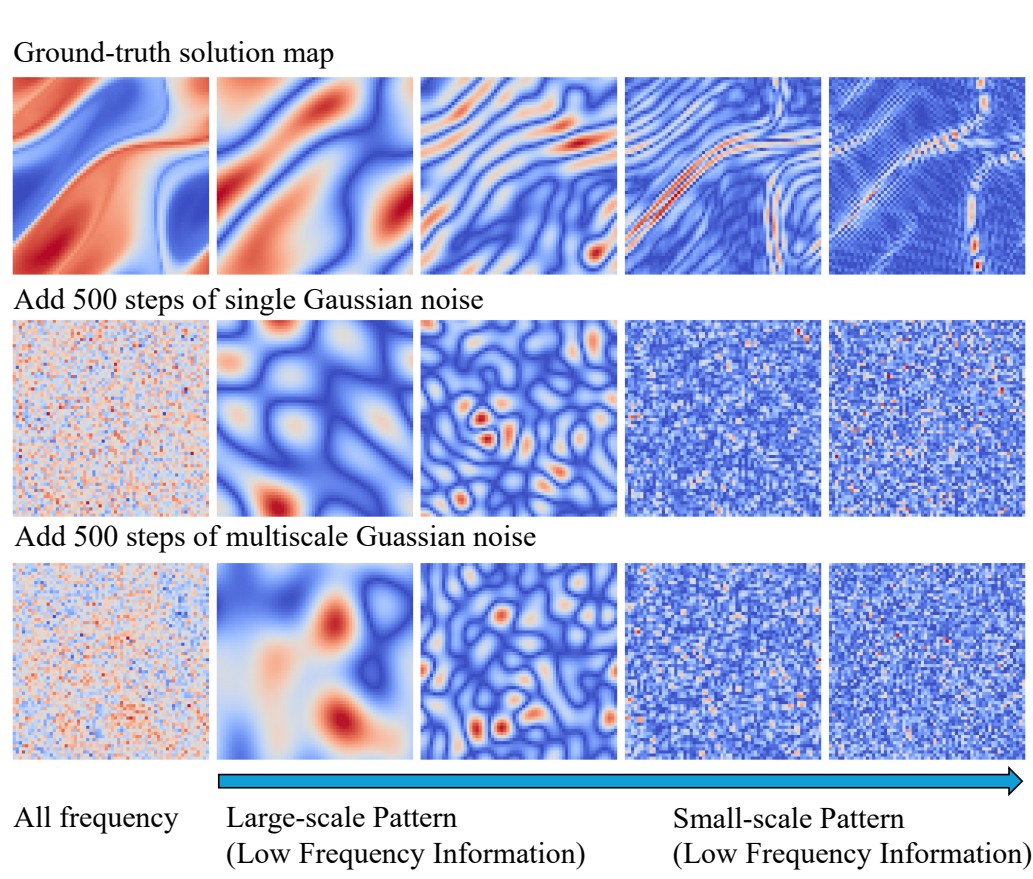

Add 500 steps of single Gaussian noise

Add 500 steps of multiscale Guassian noise

All frequency      Large-scale Pattern          Small-scale Pattern
                   (Low Frequency Information)   (Low Frequency Information)

Figure 12: Illustration of the noisy solution maps at different frequencies using Guassian noise and multi-scale noise, respectively. Guassian and multi-noise perturbations are applied to the original images 500 times each, followed by Fourier and inverse Fourier transforms to extract different frequency components(0-3, 3-7, 7-20, 20-56) based on their distance from the zero-frequency point.

