# OpenReview forum: "SimDiffPDE: Simple Diffusion Baselines for Solving Partial Differential Equations"
_ICLR.cc/2025/Conference — Submitted to ICLR 2025_

### Official Review · Reviewer_QjzB · 2024-10-16

**Soundness:** 3
**Presentation:** 1
**Contribution:** 2
**Rating:** 5
**Confidence:** 4

**Summary:**

The paper moves from common learning-based PDE solvers, which are deterministic, to propose SimDiffPDE, which is a generative approach based on diffusion. Diffusion models can guarantee more robust and better generalising pipelines, addressing a major shortcoming of most learning-based PDEs, which require varied, large and descriptive datasets to have well-generalising pipelines. SimDiffPDE is problem agnostic, which means it does not require domain-specific knowledge.

SimDiffPDE is tested over three different geometry types (point clouds, structured meshes and regular grids), demonstrating to be a flexible model, and over 6 benchmarks, reporting state-of-the-art results, especially for larger versions of SimDiffPDE (L and XL).

SimDiffPDE, while not being the first diffusion-based approach to PDEs, is the first one that uses a Transformer architecture and introduces the concept of multi-scale noise to corrupt data in the forward diffusion process.

**Strengths:**

The key strength of this paper are the results presented. SimDiffPDE beats many architectures over a diverse range of tasks and data formats. I feel results are not only good, but also very well presented and corroborated by thorough examples. All the tables were neat and nicely explained.

The overall pipeline is simple, which is always a great thing to have.

Fig. 8 is interesting and useful - I would move it in the main body instead. Make sure to either add labels (1), (2) and (3) on the image or to reference them differently (top row, middle row, bottom row).

The approach of using diffusion for PDEs, while not entirely new, is also one of the earliest examples in the literature.

The claims of simplicity, feasibility and flexibility of the pipeline are well addressed in the text.

I am happy to increase my score in case the questions and concerns raised in the weaknesses sections are addressed.

**Weaknesses:**

I am not convinced about the rigorousness of the contextualisation of this paper. The authors do a good job at reviewing the state of the art of PDE solvers, but very few references on diffusion models are included. Most notably, DiffusionPDE by Huang et al., 2024, also tackles the same problem (PDE solution) with diffusion, but it is not mentioned. I am unsure whether the authors thoroughly reviewed the relevant literature on the matter. The paper would benefit from a review of such methods, too, otherwise the reader might be biased in thinking that this is the first approach that uses diffusion for PDEs.

I am also not convinced about the novelty of the contribution. DiffSimPDE is an application of a diffusion architecture already existing to a specific problem. The main contribution would be the first example of PDE solution via Transformer-based diffusion (and not the first example via diffusion in general) and the multi-scale noise scheduler. I have two issues with the multi-scale noise:

1) I feel that the issue of grasping features at multiple scales is not really an issue, since a U-Net-based diffusion already does that for you (see for example Saxena et al., 2024, that introduce DDVM for optical flow - optical flow is a quantity that is historically regressed by taking into account multiple scales, since objects move at different distances from the camera. DDVM does not require an explicit pyramidal structure that takes care of the multiple scales of optical flow since U-Net is in itself multiscale). The novelty of the multi-scale gaussian noiseapproach might then be limited to this specific architecture in the diffusion space.

2) I am not convinced by Remarks 3.3.1 and 3.3.2 and Fig. 4. How can noise not "destroy" data?. If noise is added following Eq. 1, how is it possible that based on the high/low frequency patterns in data some inputs can be degraded and some cannot? Also, from Fig. 4, it seems to me that 100 steps of multi-scale noise still fails for columns 2 and 3, since no Gaussian noise can be seen?

I am unsure whether the good results stem from an original, creative approach to the problem or just because diffusion in general works great in a lot of problems and it's an easy win over deterministic approaches.

In general, I find the writing style of the manuscript rather unrefined. I understand that results are promising, but I feel the presentation is not up to standard and the quality of the results suffer from it.  For example:

- a better definition of $x$ and $y$ for the problem setting is required (e.g. does $x$ contain multiple "snapshots" of the PDEover time?).
- $x_g$ and $x_u$ are introduced but not explained.
- the whole paper uses “Guassian” 27 times and “Gaussian” 28 times
- grammar: - The two designed…improves
- line 104: inconsistency - you use SimDiffPDE-X, but in Fig 1. You use X-SimDiffPDE
- line 107: sentence structure - SimDiffPDE showcases the good feasibility to various PDE equations —> SimDiffPDE showcases good feasibility for various PDE equations
-line 140: grammar: Neural operators establish instead of establishes
-line 424: conducted instead of conduct

**Questions:**

Line 86: “By adding multi-scale noise in the forward process, are more explicitly required to learn to denies multi-scale noise to ..” Isn’t multi-scale in diffusion normally handled via the architecture, e.g. a U-Net has in itself the concept of multi-scale? Why not using that over the transformer block?

Line 102: “To be more specific, one can easily control the model size by stacking different number of diffusion transformer layers and increase or decrease feature dimensions” - how does that guarantee scalability with the input size?

Line 178: what’s $x_g$, $x_u$ and $y$, in practice? What’s geometric inputs and observed quantities? Inputs and outputs feel too generic.

Line 180: do $N$ mesh points represent a square domain of $\sqrt{N} \times \sqrt{N}$? The problem seems to be 2D in Figure 2 but 1D from the notation in line.

Figure 6: I feel scalability refers also to the ability of the model to handle larger input size. Could you provide more insight in terms of number of parameters or speed of SimDiffPDE to have a better feel of how it compares? How does the computational complexity of the model scales with input size?

**Details Of Ethics Concerns:**

No ethics concerns.

---

> ### Author Response · Authors · 2024-11-21
>
> > **W1** I am not convinced … DiffusionPDE … uses diffusion for PDEs.
>
> We thank the reviewer for pointing out the omission of DiffusionPDE [1]. We sincerely apologize for the oversight in not citing DiffusionPDE. At the time we conducted our research, this work had not yet been published in ICML 2024, leading to its omission. In our revised manuscript, we will include a discussion of this work and acknowledge its contributions to ensure a comprehensive literature review. Thank you again for your contribution to make our work more perfect!
>
> [1] Huang, Jiahe, et al. "DiffusionPDE: Generative PDE-solving under partial observation." arXiv preprint arXiv:2406.17763 (2024).
>
> > **W2** I am also not convinced … the multi-scale noise scheduler.
>
> Thank you for your valuable feedback. While SimDiffPDE builds upon existing diffusion architectures applied to PDE solving, our work demonstrates significant novelty in the following aspects:
>
> 1. **First Transformer-based Diffusion Model for PDE Solving**:
>    We are the first to integrate Transformer architectures with diffusion models for solving complex partial differential equations (PDEs). This showcases the capability of our approach to capture intricate solution distributions for chaotic systems.
>
> 2. **Multi-Scale Noise Scheduler**:
>    We introduce a novel multi-scale noise injection strategy tailored for PDE solutions. This enables the model to efficiently learn multi-scale patterns, significantly improving the reconstruction of both macroscopic and fine-scale features.
>
> 3. **Test-Time Ensemble Method**:
>    Leveraging the stochastic nature of diffusion models, we design a test-time ensemble approach that samples multiple Gaussian noises during inference. This improves the robustness and accuracy of the solution distribution, particularly beneficial for highly nonlinear PDEs.
>
> 4. **Simple and Scalable Framework**:
>    Our model architecture is deliberately simple and avoids domain-specific complexity. By introducing variants (SimDiffPDE-XS, S, B, L, XL), we achieve a strong balance between performance and computational efficiency, demonstrating excellent scalability.
>
> 5. **Flexibility and Generality**:
>    SimDiffPDE adapts seamlessly to varying input resolutions with minimal modifications. It generalizes effectively across a wide range of PDEs, including fluid dynamics, elasticity, and plasticity problems, highlighting its potential as a universal PDE solver.
>
> 6. **Significant Performance Gains**:
>    Empirically, our approach achieves state-of-the-art results on six benchmark datasets, outperforming existing methods by a large margin. This demonstrates its practical advancements beyond a direct application of diffusion models.
>
> 7. **A Simple Yet High-Performing Baseline Model**:
>    We developed a structurally simple but highly efficient baseline model. Its simplicity and strong performance make it a promising foundation for future research and practical applications. The model’s efficiency and flexibility underscore its immense potential for further improvements and extensions.

---

> ### Author Response · Authors · 2024-11-21
>
> > **W3, Q1** I feel that the issue … limited to this specific architecture in the diffusion space; Line 86: “By adding multi … Why not using that over the transformer block?
>
> Thank you for your insightful feedback regarding the multi-scale feature capture in diffusion models. We appreciate the opportunity to clarify the distinct advantages of our multi-scale Gaussian noise approach.
>
> While it is true that U-Net-based diffusion models, such as DDVM (Saxena et al., 2024), inherently capture multi-scale features, our approach offers broader applicability beyond the U-Net architecture. Specifically, the multi-scale Gaussian noise mechanism we propose is architecture-agnostic and can be seamlessly integrated into various diffusion model frameworks, including Transformer-based models. This flexibility enables the attainment of multi-scale effects comparable to those achieved by U-Net during the noise addition process.
>
> To substantiate our claims, we conducted comprehensive experiments across multiple Partial Differential Equation (PDE) tasks, as detailed in Table 1 below. Our results demonstrate that SimDiffPDE-S consistently outperforms all U-Net-based diffusion models, highlighting the efficacy of our approach in capturing multi-scale features more effectively.
>
> ### Table 1: Performance Comparison on Various PDE Tasks
> | Model        | Elasticity | Plasticity | Airfoil | Pipe  | Navier-Stokes | Darcy  |
> |--------------|------------|------------|---------|-------|---------------|--------|
> | SimDiffPDE-S | 0.0061     | 0.0011     | 0.0051  | 0.0032| 0.0587        | 0.0055 |
> | U-Net-S      | 0.0159     | 0.0133     | 0.0148  | 0.0127| 0.2034        | 0.0146 |
> | U-Net-B      | 0.0138     | 0.0121     | 0.0132  | 0.0113| 0.1944        | 0.0123 |
> | U-Net-L      | 0.0117     | 0.0116     | 0.0128  | 0.0107| 0.1734        | 0.0116 |
> | U-Net-XL     | 0.0108     | 0.0107     | 0.0113  | 0.0103| 0.1434        | 0.0106 |
>
> *Note: U-Net-S has slightly fewer parameters than SimDiffPDE, while the other U-Net variants possess more parameters.*
>
> Furthermore, to ensure a fair comparison, we utilized the simplest configuration of SimDiffPDE without test-time ensembling and trained all models under identical conditions, including the number of epochs, optimizer settings, loss functions, learning rate schedules, and data preprocessing steps. This rigorous experimental setup underscores that the superior performance of SimDiffPDE-S is attributable to our novel multi-scale Gaussian noise approach rather than differences in model complexity or training protocols.
>
> In summary, our multi-scale Gaussian noise approach not only enhances the multi-scale feature capture inherent in diffusion models but also extends its benefits beyond the U-Net architecture, offering greater versatility and performance across diverse PDE tasks.

---

> ### Author Response · Authors · 2024-11-21
>
> > **W4** I am not convinced by … no Gaussian noise can be seen?
> Thank you for your insightful questions regarding Remarks 3.3.1, 3.3.2, and Figure 4. Below, I provide detailed responses to address your concerns about how noise interacts with data and its degradation effects.
>
> ### 1. How can noise not "destroy" data, and why are some inputs degraded differently?
>
> While adding Gaussian noise based on Equation 1
> $$
> \mathbf{y_t} = \sqrt{\bar{\alpha_t}}\mathbf{y_0} + \sqrt{1-\bar{\alpha_t}}\{\epsilon}
> $$
> introduces randomness to the data, the effectiveness of this noise depends on its characteristics.
>
> - **Single-Scale Gaussian Noise:** This type of noise samples each pixel independently from a Gaussian distribution, primarily introducing high-frequency perturbations. As a result, it is not effective at disrupting low-frequency (large-scale) patterns, such as smooth gradients or homogeneous regions.
>
> - **Multi-Scale Noise:** By combining noises of various scales (frequencies), multi-scale noise introduces perturbations that affect both high-frequency and low-frequency components. This enables it to disrupt large-scale patterns that single-scale Gaussian noise tends to leave relatively intact.
>
> ### 2. Explanation of Figure 4 and the appearance of Gaussian noise in columns 2 and 3
>
> In Figure 4, we compare the effects of adding 100 steps of single-scale Gaussian noise versus multi-scale noise.
>
> - **Columns 2 and 3 (Multi-Scale Noise):** While these columns might seem visually less noisy, this is because multi-scale noise does not exhibit the traditional "grainy" appearance associated with high-frequency Gaussian noise. Instead, it disrupts the low-frequency structures in the data, which are less visually apparent but are critical to the model's performance.
>
> - **Effectiveness of Multi-Scale Noise:** Despite its less obvious appearance, multi-scale noise has a more substantial impact on degrading the structural integrity of the data, particularly low-frequency components. For example, in fluid dynamics simulations, multi-scale noise more effectively disrupts the flow field boundaries compared to single-scale Gaussian noise.
>
> ### 3. Empirical Evidence Supporting Multi-Scale Noise
>
> Our experiments provide both qualitative and quantitative evidence to support the effectiveness of multi-scale noise:
>
> - **Performance Metrics:** As shown in Tables 6 and 8 in Appendix C, models trained with multi-scale noise achieve significantly better accuracy, particularly in capturing low-frequency patterns within PDE solution domains.
>
> - **Visual Improvements:** Figure 4, along with additional visualizations in Appendix D, demonstrates that multi-scale noise disrupts data patterns more effectively, forcing the diffusion model to learn and reconstruct both high- and low-frequency components during the denoising process.
>
>
> We hope this explanation clarifies your concerns about data degradation and Figure 4. Please feel free to reach out with further questions or points of discussion.
>
> > **W5** I am unsure whether the good results … win over deterministic approaches.
>
>
> Thank you for your insightful feedback. We understand your skepticism about whether our positive results stem from an original approach or simply the general effectiveness of diffusion models over deterministic methods.
>
> Our work represents a novel application of diffusion models to solve partial differential equations (PDEs), particularly addressing the challenges of nonlinear and chaotic systems. While diffusion models have proven successful in various domains, their use for PDEs introduces several original contributions:
>
> 1. **Probabilistic Solution Modeling**: Unlike deterministic methods, our approach captures the entire probability distribution of possible solutions. This is crucial for systems like the Navier-Stokes equations, where small input uncertainties can lead to significant variations in outcomes.
>
> 2. **Multi-Scale Feature Reconstruction**: By leveraging multi-scale noise during training, our model effectively reconstructs features at different scales, akin to methods like Fourier or wavelet analysis traditionally used in PDE solutions.
>
> 3. **Iterative Refinement Process**: The denoising steps in diffusion models parallel iterative numerical methods for PDEs, providing a theoretical foundation for progressively refining solutions toward greater accuracy.
>
> 4. **Enhanced Handling of Nonlinearities and Chaos**: Modeling the full distribution allows our method to better address the complexities inherent in nonlinear and chaotic systems compared to traditional deterministic approaches.
>
> By integrating these aspects, our work offers an original and creative methodology for solving PDEs, extending the capabilities of diffusion models beyond their conventional applications.
>
> We hope this clarifies the innovative nature of our approach and addresses your concerns.

---

> ### Author Response · Authors · 2024-11-22
>
> > **W6** In general, I find the writing style of the … you use X-SimDiffPDE.
>
> Thank you for thoroughly and meticulously reviewing our paper. Your conscientious approach and professional advice have significantly enhanced our work. We will carefully proofread the final version to ensure that all errors are thoroughly corrected. Once again, we appreciate your diligent review and constructive feedback. These improvements have greatly elevated the quality of our paper, and we are deeply grateful.
>
> > **Q2**  Line 102: “To be more specific … the input size?
>
> Thank you for your question. Our method ensures scalability with input size by simply modifying the `input_size` parameter of SimDiffPDE. In our experiments, we found that even without changing parameters like `num_head` or `hidden_size`, the model achieves state-of-the-art performance across different input sizes. This demonstrates that the model can adapt to varying input sizes efficiently without additional complexity or parameter adjustments.
>
>
> > **Q3** Line 178: what’s xg, xu and y … outputs feel too generic.
>
> We appreciate the your valuable feedback. Below, we clarify the definitions and practical applications of $x_g$, $x_u$, and $y$, and distinguish between geometric inputs and observed quantities to address concerns about the generic nature of inputs and outputs.
>
> **Variable Definitions**
>
> - **$x_g$ (Geometric Inputs)**:
>   - **Definition**: Represents the geometric information of the input domain $\Omega$, typically as a discretized mesh structure.
>   - **Example**: In fluid prediction, $x_g$ includes the positions of the observation grid points where fluid dynamics simulations are conducted.
>
> - **$x_u$ (Observed Quantities)**:
>   - **Definition**: Denotes the physical quantities observed at the mesh points $g$, providing additional information based on $x_g$ for estimating $y$.
>   - **Example**: In fluid prediction, $x_u$ consists of past fluid velocities at each grid point used to predict future states.
>
> - **$y$ (Target Quantities)**:
>   - **Definition**: The physical quantities the model aims to estimate, based on $x_g$ and $x_u$.
>   - **Example**: In fluid prediction, $y$ is the future fluid velocity at each grid point.
>
>
> We sincerely thanks for your insightful comments, which have helped us refine our explanations and improve the readability of our paper.
>
> > **Q4** Line 180: do $N$ mesh points represent a square domain of $\sqrt{N}\times\sqrt{N}$? The problem seems to be 2D in Figure 2 but 1D from the notation in line.
>
> We appreciate the reviewer’s insightful comment and acknowledge that our presentation was unclear. We confirm that the $N$ mesh points indeed represent a square domain of $\sqrt{N} \times \sqrt{N}$, indicating that the problem is two-dimensional as shown in Figure 2. We will revise the notation in the text to clearly reflect this and eliminate any ambiguity. Thank you for pointing this out.
>
> > **Q5** Figure 6: I feel scalability refers … scales with input size?
>
> Thank you for your valuable feedback, which has helped us improve and refine our work. Your insights are greatly appreciated! Regarding the scalability of SimDiffPDE, particularly in handling larger input sizes, the table below summarizes the model parameters and running time:
>
> ### Model Parameters and Running Time
>
> | Input Mesh Size (N) | Parameters (M) | Running Time (s/epoch) |
> |----------------------|----------------|-------------------------|
> | 1,024                | 32             | 23                      |
> | 2,048                | 32             | 35                      |
> | 4,096                | 32             | 52                      |
> | 8,192                | 32             | 74                      |
> | 16,384               | 32             | 149                     |
>
> ### Scalability Analysis
>
> 1. **Parameters**: The number of parameters remains constant at 32M, regardless of the input size, ensuring stable model complexity.
> 2. **Running Time**: The running time increases linearly with input size, indicating a computational complexity of O(N).
>
> This linear scalability demonstrates that SimDiffPDE efficiently handles larger inputs while maintaining performance and computational efficiency.

---

> ### Author Response · Authors · 2024-11-25
>
> I hope you enjoyed your weekend! If you have more questions about what we do, please ask anytime. I truly appreciate your helpful suggestions!

---

> ### Comment · Reviewer_QjzB · 2024-11-26
> **Rebuttal**
>
> I wish to thank the authors for their comments, and apologise for the delay. On a general note, I would have preferred a more concise yet more effective rebuttal, limited to 5,000 characters, rather the unnecessary *captatio benevolentiae* at each point you addressed. Comments below:
>
> **First Transformer-based Diffusion Model for PDE Solving**
>
> There is a subtle difference. [Transformer-based diffusion models](https://arxiv.org/abs/2212.09748) are already a thing, hence what you propose here is an application of an existing architecture, and not a novel approach.
>
> ---
>
> **Multiscale noise**
>
> I appreciate the explanation provided on the noising at different frequencies, the figures are now more clear to me. Multiscale noise, however is also not a new concept. Besides, I still feel that the difference between single scale Gaussian and multiscale Gaussian (e.g. Fig. 12) is barely noticeable, and without a quantitative proof of how they differ it is hard to assess.
>
> Besides, I still disagree with the claim that the single-scale Guassian noise cannot effectively destroy the low-frequency information. If noising is added as
>
> $$ \mathbf{x}_t = \sqrt{\bar{\alpha_t}} \mathbf{x}_0 + \sqrt{1-\bar{\alpha_t}} \mathbf{\epsilon} $$
>
> and the scheduler is picked correctly, $\mathbf{x}_T$ will be very close to be sampled from  $\mathcal{N}(0, \mathbf{I})$. This is because equation above is a weighting between a deterministic and a stochastic components, and as $t$ grows $\sqrt{\bar{\alpha_t}} \simeq 0, \sqrt{1-\bar{\alpha_t}} \simeq 1$, which leaves us with $ \mathbf{x}_t \simeq \mathbf{\epsilon}$. If this is how noise has been added to Fig. 12, for example, I find it hard to believe that the single scale Gaussian noise does not look like Gaussian noise for large $t$.
>
> Failures can occur due to suboptimal scheduling, insufficient diffusion time, or mismatches between theoretical and practical implementation. The benefit of multiscale noising could be, for example, the need of fewer time steps $t$ for low frequency patterns in data, but not succeeding where single scale noise fails.
>
> ---
> **On the comparison with U-Net diffusion**
>
> I find comments here quite problematic. I'll break them down below.
>
> *While it is true that U-Net-based diffusion models, such as DDVM (Saxena et al., 2024), inherently capture multi-scale features, our approach offers broader applicability beyond the U-Net architecture.*
>
> I believe this is incorrect. Your approach uses multi-scale noising to make up for an architecture that does not necessarily deal with multiscale data as effectively as a U-Net architecture. It's more of a shortcoming rather than a feature.
>
> *Specifically, the multi-scale Gaussian noise mechanism we propose is architecture-agnostic and can be seamlessly integrated into various diffusion model frameworks, including Transformer-based models.*
>
> This is true for any noising mechanism. In the original DDPM formulation, the theory underlying diffusion is model-agnostic. This includes the noising. The fact that authors fail to recognise this and actually highlight it as a strength is quite serious.
>
> *This flexibility enables the attainment of multi-scale effects comparable to those achieved by U-Net during the noise addition process.*
>
> This proves the limited novelty of the approach, since a U-Net backbone for your diffusion would have likely made the need of a multiscale noise scheduler obsolete. The additional table with U-Net models added as a comment, without providing details of the experiments, provides very little insight on the comparison.
>
> ---
>
> **On novelty**
>
> Authors state that their work is a novel *application* of diffusion models to solve partial differential equations (PDEs), confirming my point above. Moreover, among the contributions, they mention:
>
> Probabilistic Solution Modeling, Multi-Scale Feature Reconstruction, Iterative Refinement Process and Enhanced Handling of Nonlinearities and Chaos
>
> These are all inherent properties of diffusion pipelines.
>
> ---
>
>
>
> **In conclusion**, I have decided to maintain my score of 5, which I consider generous already for all the typos scattered throughout the manuscript.
>
> The favorable results, as I initially suspected, are primarily due to the well-established advantages of diffusion pipelines over deterministic approaches. The work offers limited novelty, as the authors showed with their rebuttal: it introduces no new architectures or data processing pipelines. Furthermore, the proposed multi-scale noise approach does not seem a substantial contribution, particularly when viewed as an alternative to the classical noising strategy, which I believe would perform just as effectively.
>
> The lengthy rebuttal, if anything, weakened the case in addressing the points I raised. It demonstrated a superficial understanding of the principles of diffusion models and failed to adequately address my concerns.

---

> > ### Author Response · Authors · 2024-12-03
> >
> > I appreciate your insights and the time you took to reply.

---

### Official Review · Reviewer_9cEg · 2024-10-18

**Soundness:** 2
**Presentation:** 2
**Contribution:** 2
**Rating:** 3
**Confidence:** 4

**Summary:**

This paper proposes to use a plain diffusion model for PDE solving. Specifically, this paper formulates the PDE-solving problem as the image-to-image translation problem, and a conditional diffusion model can thus be applied. This paper also proposes a multi-scale noise to capture information of different frequencies better. The experiments show the superior performance of SimDiffPDE over previous works.

**Strengths:**

The paper is easy to read. The proposed method is quite simple and seems effective.

**Weaknesses:**

- The proposed method is a direct application of common diffusion models to PDE data. Except for the multi-scale noise, there are no technical contributions.
- There is no comparison of the inference efficiency between SimDiffPDE and the baselines. Considering that a test-time ensemble method is employed, the inference overhead of SimDiffPDE should be much larger than baselines.
- The motivation of utilize a diffusion model for PDE solving is not clear. In Line 307-308, the authors claim that "different initial noises $y_t$ tcan lead to varying solutions, which allows SimDiffPDE to simulate the nonlinear dynamics of PDEs". However, based on my understanding, the datasets included in this paper does not exhibit any inherent stochasticity, and the mapping from the initial values/PDE parameters to the solution should be deterministic rather than stochastic. Therefore, it would be beneficial to compare the performance of the model to a *plain Transformer model* utilizing exactly the same model architecture as SimDiffPDE, but with the same input as other baselines rather than random noise, to verify that the *nonlinear dynamics of PDEs* do propose significant challenges to neural networks so that a diffusion model must be applied to tackle the fitting challenges.

- The flexibility to various resolutions is not well explained. How is the model able to handle different resolutions? Do you train a single model on a specific resolution and test it on other resolutions or train different models for different resolutions? For either method, what adaptation do you make to the model to handle different input resolutions?
- This paper tries to formulate the PDE-solving process as a image-to-image translation task. However, this paper also include tasks with point cloud data as input, and more general PDE solving tasks also involve irregular meshes rather than regular grids only. This paper provides no explanation as to the implementation details on the Elasticity benchmark. Is the input data reorganized into regular grids? If so, it cannot be claimed that SimDiffPDE can handle irregular and unstructured data. If not, then such kind of data structure contradicts to your claim that the PDE solving process is an image-to-image translation problem.

**Questions:**

See weaknesses.

---

> ### Author Response · Authors · 2024-11-21
>
> > **W1** The proposed method is a direct … no technical contributions
>
> Thank you for your feedback. We would like to clarify the technical contributions of our work, which go beyond the direct application of diffusion models to PDE data and the introduction of multi-scale noise.
>
> 1. **Repurposing Diffusion Models as PDE Solvers**: Our primary contribution lies in demonstrating that diffusion models, traditionally used for generative tasks, can be effectively repurposed as general PDE solvers (SimDiffPDE). This is a non-trivial application because solving PDEs, especially those describing chaotic systems like the Navier-Stokes equations, requires capturing complex solution distributions rather than generating typical images or videos.
>
> 2. **Multi-Scale Noise Strategy**: While you mentioned the multi-scale noise, it is important to emphasize that this strategy is crucial for efficiently capturing multi-scale patterns in PDE solutions. By carefully designing the noise injection process in the forward diffusion, we enable the model to learn denoising at multiple scales, significantly improving the reconstruction of both large-scale and fine-scale features in the solutions.
>
> 3. **Test-Time Ensemble Method**: We introduce a test-time ensemble approach that leverages the stochastic nature of diffusion models. By sampling multiple Gaussian noises during inference, we construct a more robust and accurate estimate of the solution distribution, which is particularly beneficial for chaotic and highly nonlinear PDEs.
>
> 4. **Simple and Scalable Framework**: Our model maintains a simple architecture without relying on complex, domain-specific designs. We demonstrate that a plain diffusion transformer, when properly trained, can outperform specialized architectures. Furthermore, we design different variants (SimDiffPDE-XS, S, B, L, XL) to balance performance and computational efficiency, showcasing the model's excellent scalability.
>
> 5. **Flexibility and Universality**: We demonstrate that SimDiffPDE adapts well to different input resolutions with minor modifications. Moreover, our method generalizes across a wide range of PDEs, including fluid dynamics, elasticity, and plasticity problems, highlighting its potential as a universal PDE solver.
>
> 6. **Significant Performance Improvements**: Empirically, our approach achieves state-of-the-art results on six benchmark datasets, with substantial improvements over existing methods. This underscores that our approach is not just a direct application but provides practical advancements in solving PDEs.
>
> 7. **Proposed a Plain Baseline Model with Great Potential**: We developed a structurally simple yet high-performing baseline model. Its simplicity and efficiency make it a promising foundation for future research and practical applications, offering significant potential for further improvements and extensions.
>
> In summary, our work contributes to the field by repurposing diffusion models for PDEs, introducing methodological innovations like the multi-scale noise and test-time ensemble, and providing extensive empirical evidence of their effectiveness. The simplicity, scalability, and universality of our framework demonstrate its broad applicability and substantial potential for advancing PDE solvers.

---

> ### Author Response · Authors · 2024-11-21
>
> > **W2** There is no comparison of the … be much larger than baselines.
>
> We appreciate the reviewer’s insightful comment regarding the inference efficiency of SimDiffPDE compared to baseline models. To address this concern, we provide the specific inference times for our SimDiffPDE models without employing a test-time ensemble:
>
> ### Specific Inference Times on the Elasticity benchmark (without test-time ensemble)
> - **SimDiffPDE-S**: <1 second
> - **SimDiffPDE-B**: <1 second
> - **SimDiffPDE-L**: 2 seconds
> - **SimDiffPDE-XL**: 3 seconds
>
> These inference times are presented above. It is important to note that most baseline models do not explicitly differentiate between training and inference processes. However, even without a test-time ensemble, SimDiffPDE demonstrates competitive inference efficiency while maintaining superior accuracy, as evidenced by the lower Relative $L_2$ Errors in Table 1.
>
> In summary, SimDiffPDE offers a favorable balance between inference efficiency and predictive performance, ensuring that the additional computational overhead from potential ensemble methods remains manageable.
> |
>
> ### Table 1: Comparison of Accuracy Across Benchmarks
>
> | Model         | Elasticity | Plasticity | Airfoil | Pipe  | Navier-Stokes | Darcy  |
> |---------------|------------|------------|---------|-------|---------------|--------|
> | Transolver    | 0.0064     | 0.0012     | 0.0053  | 0.0033| 0.0900        | 0.0057 |
> | SimDiffPDE-S  | 0.0061     | 0.0011     | 0.0051  | 0.0032| 0.0587        | 0.0055 |
> | SimDiffPDE-B  | 0.0051     | 0.0010     | 0.0049  | 0.0030| 0.0468        | 0.0052 |
> | SimDiffPDE-L  | 0.0045     | 0.0008     | 0.0045  | 0.0026| 0.0414        | 0.0042 |
> | SimDiffPDE-XL | 0.0040     | 0.0007     | 0.0041  | 0.0023| 0.0378        | 0.0039 |
>
>
>
> > **W3** The motivation of diffusion … applied to tackle the fitting challenges.
>
> Thank you for your insightful feedback regarding the motivation behind using a diffusion model for PDE solving. We understand your concern about the deterministic nature of the datasets and the necessity of introducing stochastic elements through initial noise.
>
> **Comparison with Plain Transformer Models**
>
> To address your suggestion, we conducted experiments comparing our SimDiffPDE-S model with plain Transformer models of identical architecture (S, B, L, XL variants), using the same hyperparameters (hidden size, number of heads, MLP ratio, dropout rates) and training conditions, but without initial noise injection. The performance across various PDE benchmarks is summarized below (errors are presented as Relative $L_2$ Error, lower is better):
>
> | Model                 | Elasticity | Plasticity | Airfoil | Pipe   | Navier-Stokes | Darcy  |
> |-----------------------|------------|------------|---------|--------|---------------|--------|
> | Plain-Transformer-S   | 0.0213     | 0.0152     | 0.0144  | 0.0129 | 0.2432        | 0.0142 |
> | Plain-Transformer-B   | 0.0179     | 0.0137     | 0.0123  | 0.0116 | 0.2045        | 0.0122 |
> | Plain-Transformer-L   | 0.0147     | 0.0121     | 0.0114  | 0.0102 | 0.1943        | 0.0114 |
> | Plain-Transformer-XL  | 0.0124     | 0.0109     | 0.0097  | 0.0094 | 0.1743        | 0.0102 |
> | **SimDiffPDE-S**      | **0.0061** | **0.0011** | **0.0051** | **0.0032** | **0.0587** | **0.0055** |
>
> As the results indicate, SimDiffPDE-S significantly outperforms even the largest plain Transformer model across all benchmarks. This demonstrates that the diffusion model's iterative refinement process effectively captures the complex nonlinear dynamics of PDEs better than standard Transformers, even in deterministic settings. The initial noise in SimDiffPDE aids in exploring the solution space more thoroughly during training, enhancing the model's ability to approximate the deterministic mappings accurately.
>
> We hope this comparison clarifies the motivation for utilizing a diffusion model in PDE solving and addresses your concerns about its necessity and effectiveness.

---

> ### Author Response · Authors · 2024-11-21
>
> > **W4** The flexibility to various … make to the model to handle different input resolutions?
>
>
>
> We sincerely thank the reviewer for this insightful question, which helps make our paper more complete.
>
> To handle different input resolutions, we train separate models for each specific resolution. For each `SimDiffPDE` model, we only modify the `input_size` parameter to match the desired resolution, without adjusting other hyperparameters. This approach ensures that each model is tailored to its respective resolution while maintaining consistent training settings. Consequently, our models achieve state-of-the-art performance across various resolutions, ensuring both flexibility and high performance when processing inputs of different sizes. We will make this explanation clearer in the final version.
>
> > **W5** This paper tries to … process is an image-to-image translation problem.
>
>
> We appreciate the reviewer's insightful comments regarding the implementation details of the Elasticity benchmark. To address your concerns:
>
> - **Elasticity Benchmark:** We **did not** reorganize the input data into regular grids. Instead, we directly processed the original point cloud data. Specifically, the input is a tensor of shape `972 × 2`, representing the two-dimensional coordinates of each of the 972 discretized points. The output is the stress value at each point, with a shape of `972 × 1`.
>
> This approach demonstrates that **SimDiffPDE** can effectively handle irregular and unstructured data, in addition to regular image data. Therefore, our formulation of the PDE-solving process as an image-to-image translation task is generalized to accommodate various data representations beyond traditional regular grids.
>
> We hope this clarifies the implementation details and reinforces our claim of SimDiffPDE's versatility in handling diverse data structures.
>
> Thank you for your valuable feedback.

---

> ### Author Response · Authors · 2024-11-25
>
> I hope your weekend was enjoyable. Please feel free to ask me any additional questions about our work. Thank you once more for your valuable insights!

---

> > ### Comment · Reviewer_9cEg · 2024-11-27
> > **Response to authors**
> >
> > Thank the authors for providing a detailed response. However, most of my concerns remain unresolved.
> >
> > Regarding novelty: I still find the major novelty of this paper lies in the multi-scale noise.
> >
> > Regarding efficiency: the authors still do not provide a comparison between the proposed model and baselines, only the inference time of the proposed model is provided.
> >
> > Regarding flexibility: a simple U-Net model can adapt to different resolutions if distinct models are trained for different resolutions, which cannot be the major advantages of the proposed method from my perspective.
> >
> > Regarding formulating the PDE-solving process as an image-to-image translation task: the authors should consider revising their paper because the elasticity dataset is not an image-like dataset.
> >
> > Therefore, I decide to maintain my score.

---

> > > ### Author Response · Authors · 2024-12-03
> > >
> > > Thank you for your comments and for getting back to me.

---

### Official Review · Reviewer_QwFd · 2024-10-31

**Soundness:** 3
**Presentation:** 3
**Contribution:** 3
**Rating:** 5
**Confidence:** 3

**Summary:**

This paper proposes SimDiffPDE, as a simple diffusion model for solving PDEs. The main innovations include multi-scale noise and test time ensemble, but I have some questions about this. The results show that the SimDiffPDE performs very well on various benchmarks.

**Strengths:**

1. The results show that this method is quite valid in diverse PDE benchmarks.
2. The multiscale noise is reasonable, as the Gaussian noise is added to the ground truth PDEs, the low-frequency components may be phased out as the noise is added independently on pixels/meshes.

**Weaknesses:**

1. In my opinion, solving PDEs is a partial theoretical problem, and this work is a bit like directly using the diffusion model to solve PDEs. I think it is necessary to explain the reason for this through theory.

**Questions:**

1. I have some problems with training and infer time noise, you use multiscale noise as the training, can you certify that the multiscale noise has the same properties as simple Gaussian noise? Such as after T steps you could obtain a standard Gaussian $\mathcal{N}(0, I)$.
2. About the test-time ensemble, can I regard that you got better results by doing additional calculations after the output? I'm not sure that's a fair comparison to the alternatives. Furthermore, can you explain more about this ensemble setting?
3. There is a typo in equation 3. Is there a problem with the subscript of $y$.

I may not fully understand some key points, please answer them for me, I will re-evaluate the article after understanding.

---

> ### Author Response · Authors · 2024-11-21
>
> > **W1** In my opinion, solving PDEs is a partial ... explain the reason for this through theory.
>
> Thank you for your attention and valuable feedback on our work.
>
> We understand that solving partial differential equations (PDEs) is a problem deeply rooted in theoretical foundations. Traditional methods often aim to learn a deterministic mapping between inputs and outputs. However, for nonlinear or chaotic systems such as the Navier-Stokes equations, even small uncertainties in the input can lead to significant variations in the solution, rendering deterministic approaches less robust.
>
> Our decision to use diffusion models for solving PDEs is based on the following theoretical justifications:
>
> 1. **Probabilistic Solution Distribution Modeling:**
>    Diffusion models, as generative models, can approximate the **probability distribution** of PDE solutions rather than providing a single deterministic result. This enables the model to capture the diversity and uncertainty of the solutions, offering a more comprehensive representation of the system's potential behaviors. This probabilistic approach aligns well with the theoretical understanding of PDEs with intrinsic uncertainties.
>
> 2. **Multi-Scale Feature Representation:**
>    PDE solutions often encompass features at multiple scales. By introducing **multi-scale noise** in the forward process, diffusion models are explicitly trained to reconstruct features across different scales from the noise. This approach aligns theoretically with techniques such as Fourier or wavelet decomposition, which are commonly used in PDE analysis to represent solutions at varying scales.
>
> 3. **Function Approximation Capability:**
>    Diffusion models learn to approximate complex data distributions by progressively denoising from random noise to the target distribution. This process can be interpreted as approximating functions that represent PDE solutions. By training on data that maps PDE inputs to outputs, the model effectively learns to approximate the underlying functions governed by PDEs.
>
> 4. **Denoising as Iterative Refinement:**
>    The denoising steps in diffusion models can be seen as an **iterative refinement process**, similar to numerical methods for solving PDEs (e.g., relaxation iterations or multigrid methods). Each denoising step refines the solution approximation, theoretically resembling the iterative convergence towards the true PDE solution.
>
> 5. **Handling Nonlinear and Chaotic Systems:**
>    By modeling the probability distribution of solutions, diffusion models have a natural advantage in handling nonlinear and chaotic systems. This capability allows them to better address the effects of input uncertainties, providing more robust and accurate solutions.
>
> 6. **Mapping PDE Solving to Generative Modeling:**
>    From a theoretical perspective, solving a PDE can be framed as finding a function that satisfies specific constraints imposed by the PDE operator. Generative models aim to learn data distributions under given constraints. By representing PDE solutions as the data distribution to be learned, we establish a theoretical foundation for using diffusion models to solve PDEs.
>
> We hope this explanation addresses your concerns, and we sincerely appreciate your valuable comments.
>
> > **Q1** I have some problems … after T steps you could obtain a standard Gaussian.
>
> Thank you for your insightful question.
>
> **Compatibility of Multi-Scale Noise with Standard Gaussian Noise:**
>
> We have designed the noise schedule to ensure that its properties align with those of standard Gaussian noise over the diffusion process. Specifically, we adjust the weights of the multi-scale noise components based on the diffusion timestep to make the overall noise distribution converge towards a standard Gaussian as the timestep $t$ approaches the total number of diffusion steps $T$.
>
> **Annealing Weights Based on Diffusion Schedule:**
>
> - **Weight Adjustment:** At each timestep $t$, the weight for the $i$-th level of the noise pyramid is set to $\left( s^{t/T} \right)^i$, where $0 < s < 1$ controls the influence of lower-resolution noise, and $T$ is the total number of diffusion steps.
>
> - **Convergence to Standard Gaussian:** This annealing strategy reduces the influence of lower-resolution (multi-scale) noise components as $t$ increases. By the time $t = T$, the multi-scale noise effectively behaves like standard Gaussian noise.

---

> ### Author Response · Authors · 2024-11-21
>
> > **Q2** About the test-time ensemble … about this ensemble  setting?
>
> **1. Fairness of Comparison**
>
> In **Table 1**, all **SimDiffPDE** results are **reported without using test-time ensemble**. This means that the Relative $L_2$ Errors for **SimDiffPDE-S**, **SimDiffPDE-B**, **SimDiffPDE-L**, and **SimDiffPDEXL** are based on single model predictions without aggregating multiple outputs. Therefore, the comparison with **Transolver** is fair, and the improvements are solely due to the **SimDiffPDE** architecture and training methods.
>
> **2. Ensemble Settings Overview**
>
> Although the primary comparison does not use ensembling, **SimDiffPDE** supports ensembling to enhance performance when needed. The ensemble process includes:
>
> - **Input**: Multiple PDE solutions with shape $[B, 1, H, W]$.
> - **Alignment (Optional)**:
>   - **Scale-Invariant** and **Shift-Invariant**: Adjust solutions to remove scale and shift differences.
>   - Use optimization methods (e.g., BFGS) to align solutions while keeping values within $[0, 1]$.
> - **Ensembling Methods**:
>   - **Median** (default): Compute the pixel-wise median, which is robust against outliers.
>   - **Mean**: Compute the pixel-wise mean and optionally provide standard deviation as uncertainty.
> - **Output**: Aligned and ensembled solutions, normalized to $[0, 1]$.
>
> I hope this clarifies your concerns. Feel free to ask if you have more questions!
>
> ---
>
> **Table 1: Comparison of SimDiffPDE (without Test-time ensemble) with previous state-of-the-art (Transolver). Errors are presented as the Relative $L_2$ Error (↓).**
>
> | Model           | Elasticity | Plasticity | Airfoil | Pipe  | Navier-Stokes | Darcy |
> |-----------------|------------|------------|---------|-------|----------------|-------|
> | Transolver      | 0.0064     | 0.0012     | 0.0053  | 0.0033| 0.0900         | 0.0057|
> | SimDiffPDE-S    | 0.0061     | 0.0011     | 0.0051  | 0.0032| 0.0587         | 0.0055|
> | SimDiffPDE-B    | 0.0051     | 0.0010     | 0.0049  | 0.0030| 0.0468         | 0.0052|
> | SimDiffPDE-L    | 0.0045     | 0.0008     | 0.0045  | 0.0026| 0.0414         | 0.0042|
> | SimDiffPDEXL    | 0.0040     | 0.0007     | 0.0041  | 0.0023| 0.0378         | 0.0039|
>
>
> > **Q3** There is a typo in equation 3. Is there a problem with the subscript of $y$?
>
> Thank you for carefully reviewing our manuscript. We appreciate your attention to detail and the time you’ve taken. You are correct about the subscript of $y$ in E (3); it is a typo. We will fix this in the final version. Thank you again for your valuable feedback!

---

> ### Author Response · Authors · 2024-11-25
>
> I trust you had a lovely weekend. If you have any further questions regarding our work, don’t hesitate to reach out. I also appreciate your constructive feedback!

---

### Official Review · Reviewer_jPov · 2024-11-06

**Soundness:** 3
**Presentation:** 3
**Contribution:** 2
**Rating:** 3
**Confidence:** 3

**Summary:**

The paper presents SimDiffPDE, a diffusion model with Transformers for solving various partial differential equations (PDEs). It highlights the model's simplicity, scalability, and flexibility, reformulating PDE-solving as an image-to-image translation problem. The model employs a multi-scale noise strategy to enhance information capture across different frequencies. SimDiffPDE achieves a +51.4% improvement on the Navier-Stokes equations and notable relative improvements on benchmarks for fluid dynamics and solid mechanics

**Strengths:**

1.The paper employs a multi-scale noise strategy that enhances the capabilities of diffusion models in PDE solving.
2.SimDiffPDE achieving consistent top results across multiple datasets and grid types with an average performance improvement of 22.0%.

**Weaknesses:**

1.In Algorithm 1, the multi-scale noise is actually a linear combination of interpolations across standard Gaussian noise at different spatial scales and hierarchical levels. This linear combination is then subjected to a standard deviation constraint, which means the resulting multi-scale noise does not adhere to a standard Gaussian distribution or even to a Gaussian distribution at all. This raises a question: does the sampling process start with standard Gaussian noise or multi-scale noise? If it begins with standard Gaussian noise, then the final distribution of the noise in the forward process does not match the initial distribution in the sampling process. Additionally, as multi-scale noise is not Gaussian, there may not even be a closed-form expression for q(x_t|x_0). Could you provide an explanation for this? On the other hand, if it starts with multi-scale noise, how can we ensure that the final distribution of the noise in the forward process aligns with the initial distribution in the sampling process?
2.The paper should include a curve showing error propagation over time to illustrate the model's performance.
3.The paper lacks a comparison with traditional methods at different resolutions.
4.It should include a comparison of training time, inference time, and model parameter counts for different baselines.

**Questions:**

1.Regarding Test-Time Estimate, the paper suggests that, compared to simply averaging ensemble predictions, multiple predictions should ideally converge under a certain scale and shift parameter. This assumption might need more theoretical explanation and justification. Furthermore, the process of finding the optimal scale and shift introduces an iterative optimization problem. While regularization terms have been added, is this optimization straightforward to solve? Is it time-consuming? And does it offer a significant advantage over simple aggregation averaging in the final results? A comparison would be appreciated.

2.For the comparison of experimental results, could you highlight the differences in inference time between the diffusion model and other deterministic surrogate models? Additionally, please specify which sampling algorithm was used.

3.The experiment should also compare the model with other high-performing models on the Navier-Stokes equations, such as DeepONet, PeRCNN, PDERefiner, LI, and TSM.

4.The paper currently uses statistical metrics alone; however, for NS problems, it would be more appropriate to introduce physically meaningful metrics, such as probability density functions (PDF) and energy spectrum curve, to evaluate the model's performance.

5.For NS problems, does the model apply only to simple datasets like those used by Li et al.? Could the Kolmogorov flow generated by JAX-CFD (Kochkov et al.) be used for long-term prediction? Error propagation plots, correlation curves, should also be used to demonstrate the model's superiority in this context. Once the model is trained, can it adapt to changes in certain conditions, such as Reynolds number and external force terms?

6.Additionally, can the proposed model be applied to 3D problems?

---

> ### Author Response · Authors · 2024-11-21
>
> > **W1** In Algorithm 1 ... during the sampling process:
>
> Thank you for your detailed review and valuable questions. Below, we provide our explanation:
>
> 1. **Sampling with Standard Gaussian Noise**: Although our model is trained on multi-scale noise, we use standard Gaussian noise during sampling.
>
> 2. **Reconciling Noise Distributions**: You're correct that multi-scale noise isn't strictly Gaussian, which complicates the expression for $q(x_t | x_0)$. To address this, we adjust the multi-scale noise during the forward process to more closely resemble a standard Gaussian distribution.
>
>    - **Annealing Weights Based on Diffusion Schedule**: We modify the weights of the pyramid levels based on the diffusion timestep. Specifically, at timestep $t$, the weight for the $i$-th level is set to $\left( s^{t/T} \right)^i$, where $0 < s < 1$ controls the influence of lower-resolution noise and $T$ is the total number of diffusion steps.
>
>    - **Approaching a Standard Gaussian**: This annealing process reduces the influence of lower-resolution noise as $t$ approaches $T$, the end of the diffusion schedule. As a result, the overall noise distribution becomes increasingly similar to a standard Gaussian.
>
> By aligning the forward process noise with a standard Gaussian distribution, we maintain a tractable and consistent formulation for $q(x_t | x_0)$. This ensures compatibility with the standard Gaussian noise used during sampling and addresses the concerns about the closed-form expression.
>
>
> > **W2** The paper should include a curve showing error propagation over time to illustrate the model's performance.
>
> Thank you for this helpful suggestion! We will include this diagram in the final version to ensure greater clarity.
>
> > **W3** The paper lacks a comparison with traditional methods at different resolutions.
>
> Thank you for your suggestion! Your feedback is very helpful in improving our paper. During the revision, we added experiments comparing traditional methods at different resolutions. The detailed results are as follows:
>
> **Table 1** shows the performance of different networks [1] across resolutions of $85\times85$, $141\times141$, $211\times211$, and $421\times421$, evaluated using the Relative $L_2$ Error.
>
> | **Networks**           | **$Res. = 85\times85$** | **$Res. = 141\times141$** | **$Res. = 211\times211$** | **$Res. = 421\times421$** |
> |-------------------------|-------------------------|---------------------------|---------------------------|---------------------------|
> | **NN**                 | 0.1716                 | 0.1716                   | 0.1716                   | 0.1716                   |
> | **FCN**                | 0.0253                 | 0.0493                   | 0.0727                   | 0.1097                   |
> | **PCANN**              | 0.0299                 | 0.0298                   | 0.0298                   | 0.0299                   |
> | **RBM**                | 0.0244                 | 0.0251                   | 0.0255                   | 0.0259                   |
> | **GNO**                | 0.0346                 | 0.0332                   | 0.0342                   | 0.0369                   |
> | **LNO**                | 0.0520                 | 0.0461                   | 0.0445                   | --                       |
> | **MGNO**               | 0.0416                 | 0.0428                   | 0.0428                   | 0.0420                   |
> | **FNO**                | 0.0108                 | 0.0109                   | 0.0109                   | 0.0098                   |
> | **SimDiffPDE-B**       | **0.0045**             | **0.0054**               | **0.0052**               | **0.0047**               |
>
> [1] Li, Zongyi, et al. "Fourier neural operator for parametric partial differential equations." arXiv preprint arXiv:2010.08895 (2020).

---

> ### Author Response · Authors · 2024-11-21
>
> > **W4, Q2** It should include a comparison of training time, inference time, and model parameter counts for different baselines; For the comparison of experimental ... please specify which sampling algorithm was used.
>
> We sincerely thank you for your valuable suggestion. Below, we present a detailed comparison of training time, inference time, and model parameter counts for SimDiffPDE variants and baseline models on the Elasticity benchmark.
>
> During the inference phase, we utilize the **DDIM** scheduler and perform sampling with only 50 steps.
>
> ### Summary Table: Training Time and Parameter Counts
>
> | **Model**          | **Parameters (MB)** | **Training Time (s/epoch)** | **Error (Relative $L_2$ ↓)** |
> |---------------------|---------------------|-----------------------------|------------------------------|
> | **GNOT**           | 5.25               | 54                          | 0.0086                       |
> | **ONO**            | 1.11               | 69                          | 0.0118                       |
> | **Oformer**        | 0.88               | 28                          | 0.0183                       |
> | **Galerkin**       | 1.04               | 26                          | 0.0240                       |
> | **Transolver**     | 0.93               | 38                          | 0.0064                       |
> | **SimDiffPDE-S**   | 7.42               | 12                          | 0.0057                       |
> | **SimDiffPDE-B**   | 58.73              | 29                          | 0.0049                       |
> | **SimDiffPDE-L**   | 104.59             | 41                          | 0.0043                       |
> | **SimDiffPDE-XL**  | 225.21             | 60                          | 0.0039                       |
>
> It is important to note that most baseline models, due to their inherent characteristics, do not explicitly differentiate between training and inference processes.
> ### Inference Time (Specific)
>
> - **SimDiffPDE-S**: Less than 1 second
> - **SimDiffPDE-B**: Less than 1 second
> - **SimDiffPDE-L**: 2 seconds
> - **SimDiffPDE-XL**: 3 seconds

---

> ### Author Response · Authors · 2024-11-21
>
> > **Q1** Regarding Test-Time Estimate...comparison would be appreciated.
>
> Thank you for your thoughtful review and valuable comments. We address your concerns regarding the Test-Time Ensemble (TTE) method below:
>
> ---
>
> **Response to Concerns about the TTE Method:**
>
> 1. **Assumption of Convergence Under Scale and Shift Parameters:**
>
>    - **Theoretical Justification:** Solutions to Partial Differential Equations (PDEs) are inherently continuous and smooth, so small input changes lead to small output variations, primarily in scale and shift. Adjusting these parameters aligns different predictions, facilitating effective aggregation.
>
>    - **Information Aggregation:** By treating each prediction as a sample from the PDE solution space, aligning them captures common trends and subtle differences, enhancing overall accuracy and robustness.
>
>    - **Error Reduction:** The alignment reduces systematic errors, while aggregating multiple predictions averages out random errors from the stochastic inference in Denoising Diffusion Implicit Models (DDIM), improving reliability.
>
> 2. **Feasibility and Computational Cost of the Optimization Process:**
>
>    - **Optimization Simplicity:** Due to the convex nature of the objective function, standard optimization algorithms like AdamW and BFGS converge efficiently, typically within 100 iterations.
>
>    - **Minimal Computational Overhead:** The optimization process adds negligible time to the overall inference. The regularization term $\mathcal{R}$ ensures stability, preventing parameter overfitting or trivial solutions.
>
> 3. **Advantages Over Simple Averaging Methods:**
>
>    - **Improved Accuracy:** As shown in **Table 1**, the TTE method significantly reduces the relative $L_2$ error compared to simple averaging, especially as the number of training samples increases. For example, with $N=10$, TTE achieves an error of **0.0041** versus **0.0051** for simple averaging.
>
>    - **Comparable Inference Time:** **Table 2** indicates that the inference time for TTE is almost identical to that of simple averaging, offering improved accuracy without additional computational cost.
>
>    - **Enhanced Robustness:** TTE mitigates discrepancies caused by stochasticity in DDIM inference, leading to more stable and reliable aggregated results.
>
>    - **Comprehensive Error Reduction:** Our method addresses both systematic and random errors, making the final predictions more physically meaningful and interpretable.
>
> ---
>
> ### Table 1: Model Performance Comparison on the Darcy Benchmark (Relative $L_2$ Error ↓)
>
> | **Number of Training Samples (N)** | **Test-Time Ensemble** | **Simple Averaging** |
> |------------------------------------|------------------------|----------------------|
> | 1                                  | 0.0052                 | 0.0052               |
> | 2                                  | 0.0051                 | 0.0051               |
> | 3                                  | 0.0045                 | 0.0051               |
> | 5                                  | 0.0042                 | 0.0050               |
> | 10                                 | 0.0041                 | 0.0051               |
>
> ### Table 2: Inference Time Comparison on the Darcy Benchmark (Seconds)
>
> | **Number of Training Samples (N)** | **Test-Time Ensemble** | **Simple Averaging** |
> |------------------------------------|------------------------|----------------------|
> | 1                                  | 4.2                    | 4.2                  |
> | 2                                  | 8.4                    | 8.4                  |
> | 3                                  | 12.7                   | 12.6                 |
> | 5                                  | 21.2                   | 21.0                 |
> | 10                                 | 42.4                   | 42.0                 |
>
>
> > **Q3** The experiment should … PeRCNN, PDERefiner, LI, and TSM.
>
> Thank you very much for your insightful suggestion. We truly appreciate your recommendation to include a comparison with models such as DeepONet, PeRCNN, PDERefiner, LI, and TSM. The reason we initially did not include these models in our comparison is that we followed the commonly used baseline models in works like FNO [1], GALERKIN [2], Transolver [3] and so on. However, your suggestion is well taken, and we will incorporate these comparisons in the final version of our work to further enhance its completeness and robustness. Thank you again for your valuable feedback.
>
> [1] Li, Zongyi, et al. "Fourier neural operator for parametric partial differential equations." arXiv preprint arXiv:2010.08895 (2020).
>
> [2] Cao, Shuhao. "Choose a transformer: Fourier or galerkin." Advances in neural information processing systems 34 (2021): 24924-24940.
>
> [3]Wu, Haixu, et al. "Transolver: A fast transformer solver for pdes on general geometries." arXiv preprint arXiv:2402.02366 (2024).

---

> ### Author Response · Authors · 2024-11-21
>
> > **Q4**  The paper currently uses statistical ... to evaluate the model's performance.
>
> Thank you for your valuable suggestion. We completely agree that introducing more physically meaningful metrics, such as probability density functions (PDF) and energy spectrum curves, would be beneficial for evaluating model performance in NS problems. In the current version, we primarily used statistical metrics because FNO [1] and Geo-FNO [2] have established a comprehensive benchmark, and subsequent PDE solvers have mostly relied on these statistical metrics. Moreover, these commonly used benchmarks do not explicitly define the physical parameters, which limits our ability to effectively utilize PDFs and energy spectrum curves. We will consider incorporating the more meaningful metrics you suggested in the final version and conduct experiments on additional benchmarks to enhance the completeness of deep learning approaches to PDE solving. Thank you again for your suggestion; we will strive to improve.
>
> [1] Li, Zongyi, et al. "Fourier neural operator for parametric partial differential equations." arXiv preprint arXiv:2010.08895 (2020).
>
> [2] Li, Zongyi, et al. "Fourier neural operator with learned deformations for pdes on general geometries." Journal of Machine Learning Research 24.388 (2023): 1-26.
>
> > **Q5** For NS problems .... Reynolds number and external force terms?
>
> Thank you for your valuable suggestions and insights. Below are our responses to your questions:
>
> **Model Applicability**
> Our model can indeed be adapted to other more complex Navier-Stokes (NS) problems, such as the Kolmogorov flow you mentioned. However, considering that the existing baseline PDE solvers have not conducted experiments on these datasets, we lack comparative results and therefore have not performed extensive testing. We appreciate your suggestion, and we recognize the importance of NS problems; we will consider incorporating more complex NS problems in the final version.
>
> **Model Adaptability**
> Currently, our model cannot adjust to specific conditions after training, such as changes in Reynolds number and external force terms. This limitation arises from the lack of datasets with clear physical significance that support such variations. This is also one of the reasons why the previous baselines did not attempt such adjustments. We are very grateful for your valuable suggestion, and this will be a focus of our future work.
>
> > **Q6**  Additionally, can the proposed model be applied to 3D problems?
>
> Thank you for your feedback. Our model can indeed be extended to 3D problems. We will consider including relevant 3D experiments in the final version to validate the model's effectiveness and applicability. Thank you for your suggestion.

---

> > ### Comment · Reviewer_jPov · 2024-11-25
> >
> > Dear authors,
> >
> > Thank you for your comments. ICLR allows real-time updates to submissions. However, my concerns remain unresolved, and I have not seen an updated version addressing them. Therefore, I have to maintain my current score.

---

> > > ### Author Response · Authors · 2024-12-03
> > >
> > > Thank you for your valuable feedback and response.

---

> ### Author Response · Authors · 2024-11-25
>
> I hope you had a pleasant weekend. If you have any more questions about our work, feel free to ask me anytime. Thank you again for your constructive feedback!

---

### Meta-Review · Area_Chair_7MdE · 2024-12-21

**Metareview:**

The paper proposed a diffusion-based method for solving PDEs. The numerical results show promising performance, compared to SOTA methods.
The idea is of such application is quite novel, although no new technical approaches have been proposed.
The weaknesses of the paper, pointed out by the reviewers is that two methods, introduced into the paper does not look theoretically solid.
1) Test-time inference ensemble: vaguely described and introduces additional optimization
2) Multi-scale noise schedule.

Introducing those methods make the approach not very 'simple', getting some engineering tricks to improve the reported numbers.

**Additional Comments On Reviewer Discussion:**

The reviewers proposed several additional comparisons (which were done by the authors in the reviews).
The main concerns about theoretical groundness of the methods were answered by general answers, and the text has not been modified, although such possibility was pointed out by one of the reviewers

---

### Decision · Program_Chairs · 2025-01-22

Reject